# ADD-IT: TRAINING-FREE OBJECT INSERTION IN IMAGES WITH PRETRAINED DIFFUSION MODELS

**Yoad Tewel**
NVIDIA, Tel-Aviv University

**Rinon Gal**
NVIDIA, Tel-Aviv University

**Dvir Samuel**
Bar-Ilan University

**Yuval Atzmon**
NVIDIA

**Lior Wolf**
Tel-Aviv University

**Gal Chechik**
NVIDIA

Figure 1: Given an input image (left in each pair), either real (top row) or generated (mid row), along with a simple textual prompt describing an object to be added *Add-it* seamlessly adds the object to the image in a natural way. *Add-it* allows the step-by-step creation of complex scenes without the need for optimization or pre-training.

## ABSTRACT

Adding Object into images based on text instructions is a challenging task in semantic image editing, requiring a balance between preserving the original scene and seamlessly integrating the new object in a fitting location. Despite extensive efforts, existing models often struggle with this balance, particularly with finding a natural location for adding an object in complex scenes. We introduce *Add-it*, a training-free approach that extends diffusion models' attention mechanisms to incorporate information from three key sources: the scene image, the text prompt, and the generated image itself. Our weighted extended-attention mechanism maintains structural consistency and fine details while ensuring natural object placement. Without task-specific fine-tuning, *Add-it* achieves state-of-the-art results on both real and generated image insertion benchmarks, including our newly constructed "*Additing* Affordance Benchmark" for evaluating object placement plausibility, outperforming supervised methods. Human evaluations show that *Add-it* is preferred in over 80% of cases, and it also demonstrates improvements in various automated metrics.

# 1    INTRODUCTION

Adding objects to images based on textual instructions is a challenging task in image editing, with numerous applications in computer graphics, content creation and synthetic data generation. A creator may want to use text-to-image models to iteratively build a complex visual scene, while autonomous driving researchers may wish to draw pedestrians in new scenarios for training their car-perception system. Despite considerable recent research efforts on text-based editing, this particular task remains a challenge . When adding objects, one needs to preserve the appearance and structure of the original scene as closely as possible, while inserting the novel objects in a way that appears natural. To do so, one must first understand *affordance*—the deep semantic knowledge of how people and objects interact, in order to position an object in a reasonable location. For brevity, we call this task *Image Additing*.

Several studies (Hertz et al., 2022; Meng et al., 2022) tried addressing this task by leveraging modern text-to-image diffusion models. This is a natural choice since these models embody substantial knowledge about arrangements of objects in scenes and support open-world conditioning on text. While these methods perform well for various editing tasks, their success rate for adding objects is disappointingly low, failing to align with both the source image and the text prompt. In response, another set of methods took a more direct learning approach (Brooks et al., 2023; Zhang et al., 2023; Canberk et al., 2024). They trained deep models on large image editing datasets, pairing images with and without an object to add. However, these often struggle with generalization beyond their training data, falling short of the general nature of the original diffusion model itself. This typically manifests as a failure to insert the new object, the creation of visual artifacts, or more commonly – failing to insert the object in the correct place, *i.e.* struggling with affordances. Indeed, we remain far from achieving open-world object insertions from text instructions.

Here we describe an open-world, training-free method that can successfully leverage the knowledge stored in text-to-image foundation models, to naturally add objects into images. As a guiding principle, we propose that addressing the affordance challenge requires methods to carefully balance between the context of the existing scene and the instructions provided in the prompt. We achieve this by: first, extending the multi-modal attention mechanism (Esser et al., 2024) of recent T2I diffusion models to also consider tokens from a source image; and second, controlling the influence of each multi-modal attention component: the source image, the target image and the text prompt. A main contribution of this paper is a mechanism to balance these three sources of attention during generation. We also apply a structure transfer step and introduce a novel subject-guided latent blending mechanism to preserve the fine details of the source image while enabling necessary adjustments, such as shadows or reflections. Our full pipeline is shown at fig. 2. We name our method *Add-it*.

Image *Additing* methods typically face three main failure modes: neglect, appearance, and affordance. While current CLIP-based evaluation protocols can partially assess neglect and appearance, there is a lack of reliable methods for evaluating affordance. To address this gap, we introduce the "*Additing* Affordance Benchmark," where we manually annotate suitable areas for object insertion in images and propose a new protocol specifically designed to evaluate the plausibility of object placement. Additionally, we introduce a metric to capture object neglect. *Add-it* outperforms all baselines, improving affordance from 47% to 83%. We also evaluate our method on an existing benchmark (Sheynin et al., 2023) with *real images*, as well as our newly proposed Additing Benchmark for generated images. *Add-it* consistently surpasses previous methods, as reflected by CLIP-based metrics, our object inclusion metric, and human preference, where our method is favored in over 80% of cases, even against methods specifically trained for this task.

Our contributions are as follows: (i) We propose a training-free method that achieves state-of-the-art results on the task of object insertion, *significantly* outperforming previous methods, including supervised ones trained for this task. (ii) We analyze the components of attention in a modern diffusion model and introduce a novel mechanism to control their contribution, along with novel Subject Guided Latent Blending and a noise structure transfer. (iii) We introduce an affordance benchmark and a new evaluation protocol to assess the plausibility of object insertion, addressing a critical gap in current Image *Additing* evaluation methods.

# 2    RELATED WORK

**Object Placement and Insertion.** Inserting objects into images remains a core challenge in image editing. Traditional computer graphics methods often depend on manual object placement (C. Wang, 2014) or utilize synthetic data-driven approaches (Fisher et al., 2012). Early computer vision techniques employed contextual cues to predict possible object positions (Choi et al., 2012; Lin et al., 2013; Zhao et al., 2011). With advancements in deep learning, generative models have been trained to learn object placements. For example, Compositing GAN (Azadi et al., 2020) generates object composites by refining geometry and appearance, while RelaxedPlacement (Lee et al., 2022) optimizes object placement and sizing based on relationships depicted in scene graphs. OBJect3DIT (Michel et al., 2024) explores 3D-aware object insertion guided by language instructions, primarily using synthetic data. Despite their effectiveness, these methods often struggle with the complexities of real-world placement scenarios.

**Editing with Text-to-Image Diffusion Models.** The emergence of high-performing text-to-image diffusion models (Rombach et al., 2022; Saharia et al., 2022; Ramesh et al., 2022; Balaji et al., 2022; Esser et al., 2024) has paved the way for effective text-based image editing techniques. Methods like Prompt-to-Prompt (Hertz et al., 2022) modify attention maps by injecting the input caption's attention into the target caption's attention, while SDEdit (Meng et al., 2022) uses a stochastic differential equation to iteratively denoise and enhance the realism of user-provided pixel edits. For editing real images, inversion techniques (Mokady et al., 2023; Wallace et al., 2022; Pan et al., 2023; Samuel et al., 2023; Deutch et al., 2024; Huberman-Spiegelglas et al., 2023) first invert an input image to its latent noise representation using a given caption, enabling edits via methods like SDEdit or Prompt2Prompt. Cao et al. (2023) further improves real image editing using a mutual extended self-attention mechanism. Despite their effectiveness in various tasks, these methods struggle with object addition, often failing to align new objects with both the original image and the text prompt.

To improve editing performance, several methods proposed to directly fine-tune diffusion models. Imagic (Kawar et al., 2023) fine-tunes diffusion models to handle complex textual instructions, whereas Text2LIVE (Bar-Tal et al., 2022) and Blended Diffusion (Avrahami et al., 2022) blend edited regions throughout the generation. InstructPix2Pix (Brooks et al., 2023) introduced an instructable image editing model trained on a large synthetic dataset for instruction-based edits, while MagicBrush (Zhang et al., 2023) enhances this approach by fine-tuning InstructPix2Pix on a manually annotated dataset collected through an online editing tool. EmuEdit (Sheynin et al., 2023) trains a diffusion model on a large synthetic dataset to perform different editing tasks given a task embedding. EraseDraw (Canberk et al., 2024) leverages inpainting models to automatically generate high-quality training data for learning object insertion. They show that one can train models to realistically insert diverse objects into images based on language instructions.

Despite advancements in instruction-based image editing, we demonstrate that current methods still face significant challenges in accurately interpreting and executing object addition within images. In this paper, we propose a novel approach addressing the challenging task of object insertion. We show that by controlling the various attention components in the diffusion model, one can add new objects to existing images without further training or fine-tuning of the diffusion model.

## 3 METHOD

Our goal is to insert an object into a real or generated image using a simple textual prompt, ensuring the result appears natural and consistent with the source image. To achieve this, we leverage a pretrained diffusion model without any additional training or optimization. Our solution consists of three core components: (1) a weighted extended self-attention mechanism that balances information from the source image, text prompt, and target image, (2) a noising approach that preserves the source image's structure, and (3) a novel Subject-Guided Latent Blending mechanism to retain fine background details. For real images, we also introduce an inversion step, detailed below.

### 3.1 PRELIMINARIES: ATTENTION IN *MM-DiT* BLOCKS

Modern Diffusion Transformers (DiTs) models, such as SD3 (Esser et al., 2024) and FLUX (Black-Forest, 2024), process concatenated sequences of textual-prompt and image-patch tokens through unified multi-modal self-attention blocks (*MM-DiT* blocks). Specifically, FLUX has two types of

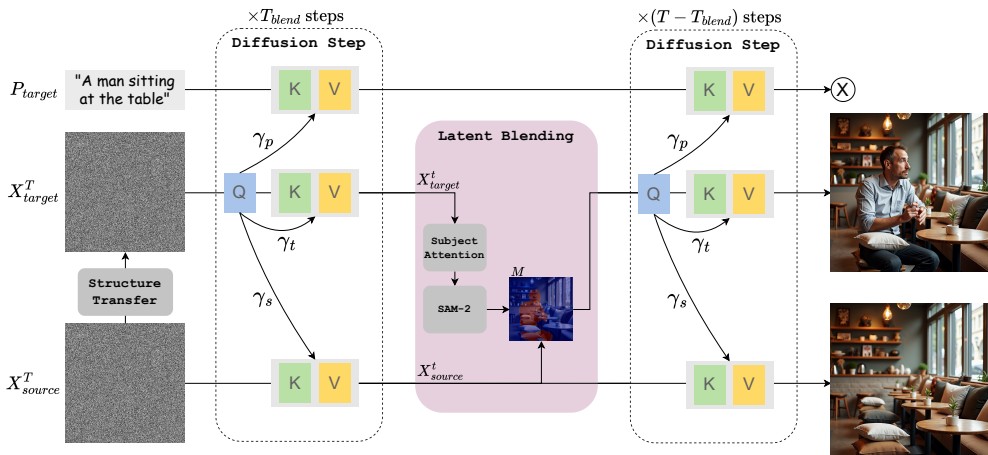

Figure 2: **Architecture outline:** Given a tuple of source noise $X_{source}^T$, target noise $X_{target}^T$, and a text prompt $P_{target}$, we first apply Structure Transfer to inject the source image's structure into the target image. We then extend the self-attention blocks so that $X_{target}^T$ pulls keys and values from both $P_{target}$ and $X_{source}^T$, with each source weighted separately. Finally, we use Subject Guided Latent Blending to retain fine details from the source image.

attention blocks: **Multi-stream** blocks which use separate projection matrices ($\boldsymbol{W}_K, \boldsymbol{W}_V, \boldsymbol{W}_Q$) for text and image tokens, and **Single-stream** blocks where the same projection matrices are used for both. Both block types compute attention on the concatenated tokens as follows:

$$A = softmax([Q_p, Q_{img}][K_p, K_{img}]^\top / \sqrt{d_k}), \quad h = A \cdot [V_p, V_{img}] \tag{1}$$

where $Q_p, Q_{img}$ are the textual-prompt and the image-patch queries, respectively. The same applies to $K$ and $V$. Notably, Flux is composed of a series of Multi-stream blocks followed by a series of Single-stream blocks.

## 3.2 WEIGHTED EXTENDED SELF-ATTENTION

Our approach builds on top of the attention mechanism in *MM-DiT* blocks. In this attention mechanism, tokens are drawn from two sources: the image patches $X_{image}$ and the textual prompt $P$. In prior attention-based diffusion architectures, it was shown that the appearance of a source image can be transferred to a target through an extended self-attention mechanism, where the new image can attend to the tokens of the source. We propose a similar extension here, by allowing the multi-modal attention to include another source — the tokens of the input image we wish to edit. More formally, we define the three sources of information as: the source image $X_{source}$, the generated image $X_{target}$ and the textual prompt describing the edit $P_{target}$. To compute the source image tokens, we simply denoise it in parallel to the target image, and concatenate its keys and values to the self-attention blocks, extending eq. (1):

$$A = softmax([Q_p, Q_{target}][K_{source}, K_p, K_{target}]^\top / \sqrt{d_k}), \quad h = A \cdot [V_{source}, V_p, V_{target}] \tag{2}$$

where $K_{source}$ and $V_{source}$ are the keys and value extracted from the source image, and $K_p$, $V_p$, $K_{target}$, $V_{target}$ are the keys and values from the prompt and target image respectively. When $X_{source}$ is a generated image, denoising it in parallel is trivial - we simply need to start denoising from the same seed that created $X_{source}$. Dealing with a real image is more complicated, and we will describe our solution in the inversion section below.

However, we notice that simply appending the keys and values of the source image to the attention blocks leads to the source image controlling the attention, which in turn leads to neglect of the edit prompt, with the final generated image being a simple copy of the source image. We explore the dynamics of this phenomenon in detail in section 5. To avoid this effect, we can re-balance the contribution of different attention components by weighting their keys. Indeed, by reducing the weight of the source image tokens, we can achieve better balance and allow for more changes. However, if this is not done carefully, then we risk upsetting the balance in the opposite fashion and

seeing alignment with the source image completely ignored. Hence, we can introduce a weighting term to each source of information, giving us the following multi-modal attention equation:

$$A = softmax([Q_p, Q_{target}][\gamma_s \cdot K_{source}, \gamma_p \cdot K_p, \gamma_t \cdot K_{target}]^\top / \sqrt{d_k})$$
$$h = A \cdot [V_{source}, V_p, V_{target}]$$

$$(3)$$

where $\gamma_s$, $\gamma_p$, $\gamma_t$ represent the weighting terms for the source image, the prompt, and the target image, respectively. In section 5 we explore the dynamics of the attention distribution across these three sources. In practice, we find that it is necessary to balance two key terms: the first is the attention distributed over the source image $A_{source} = \frac{\exp(Q_p \cdot K_{source})}{Z}$ and the second is the attention distributed over the target image, $A_{target} = \frac{\exp(\gamma \cdot Q_p \cdot K_{target})}{Z}$, where $Z$ is the softmax normalization term. To determine $\gamma$ we define the function $f(\gamma) = A_{source} - A_{target}$ and use a root-solver algorithm to find $\gamma$ such that $f(\gamma) = 0$.

## 3.3 STRUCTURE TRANSFER

The weighted extended-attention mechanism allows to balance between information from the source image and the prompt, but the added objects do not always adhere to the image context (e.g. dog is too big for the chair). We attribute this issue to different seeds dictating specific structures in the generated image, which do not always align with the source image. We show that effect in fig. 8, where images generated with the same seed produce similar objects with or without the extended attention mechanism. To address this problem, we propose to choose seeds with a structural similarity to the source image. We do so by noising the source latent $X_{source}$ to a very high noise level $t_{struct}$ with randomly sampled noise $\epsilon \sim \mathcal{N}(0, I)$ following the recitified flow denoising formula $X_t = (1 - \sigma_t)x_0 + \sigma_t\epsilon$. When $t_{struct}$ is high enough, starting the denoising process from $X_{t_{struct}}$ will result in an image with similar global structure to the source image, while still allowing for changes to image content as demonstrated in fig. 8.

## 3.4 SUBJECT GUIDED LATENT BLENDING

The combination of structure transfer and the weighted attention mechanism ensures that the target image remains consistent with the structure and appearance of the source image, though some fine details, such as textures and small background objects, may still change. Our goal is to preserve all elements of the source image not affected by the added object. To achieve this, we propose Latent Blending; A naive approach would involve identifying the pixels unaffected by the object insertion and keeping them identical to those in the source image. However, two challenges arise: First, a perfect mask is needed to separate the object from the background to avoid artifacts. Second, we aim to preserve collateral effects from the object insertion, such as shadows and reflections. To address these issues, we propose generating a rough mask of the object, which is then refined using SAM-2 (Ravi et al., 2024) to obtain a final mask $M$. We then blend (Avrahami et al., 2022) the source and target noisy latents at timestep $T_{blend}$ based on this mask.

To extract the rough object mask, we gather the self-attention maps corresponding to the token representing the object. We achieve this by multiplying the queries from the target image patches, $Q_{target}$, with the key associated with the added object token, $k_{object}$. These maps are then aggregated across specific timesteps and layers that we identified as generating the most accurate results (further details can be found in the appendix A.1. We then apply a dynamic threshold to the attention maps using the Otsu method (Otsu, 1979) to obtain a rough object mask, $M_r$. Finally, we refine this mask using the general-purpose segmentation model, SAM-2. Since SAM-2 operates on images rather than noisy latents, we first estimate an image, $X_0$, from the model's velocity prediction, $v_\theta$, using the formula $X_0 = X_{T_{blend}} + (\sigma_{T_{blend}+1} - \sigma_{T_{blend}}) \cdot v_\theta$. In addition to an input image, SAM-2 requires a localization prompt in the form of points, a bounding box, or an input mask. In our method, we provide input points, as they tend to produce the most accurate masks. To extract these localization points, we iteratively sample local maxima from the attention maps - full details of this sampling process are provided in appendix A.1. Using these input points, we generate the refined object mask, $M$. Finally, we apply a simple latent blending step at timestep $T_{blend}$, where we compute $Z_{target} = M \odot Z_{target} + (1 - M) \odot Z_{source}$. We present results with and without latent blending, along with the resulting mask $M$, in fig. 9.

|  | I-P2P | ED | MB | PbI | InfEdit | MasaCtrl | SDEdit | P2P | **Ours** |
|---|---|---|---|---|---|---|---|---|---|
| Affordance | 0.276 | 0.341 | 0.418 | 0.311 | 0.366 | 0.203 | 0.397 | 0.474 | **0.828** |

Table 1: Methods comparison based on Affordance score for the *Additing* Affordance Benchmark.

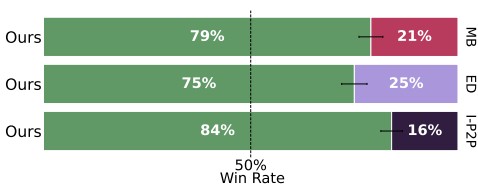

Figure 3: User Study results evaluated on the real images from the Emu Edit Benchmark.

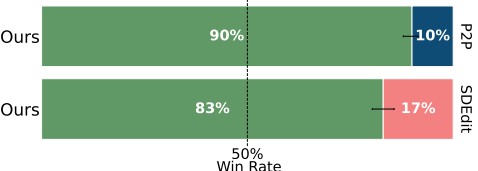

Figure 4: User Study results evaluated on the generated images from the Image *Additing* Benchmark.

## 3.5 *Additing* REAL IMAGES AND STEP-BY-STEP GENERATION

In the previous sections, we described our method for generating an edited image by drawing information from a source image within the same batch. When editing a generated image, this process is straightforward: one can save the source noise, $\epsilon_{source}$, that generated the source image and create an input batch containing both $\epsilon_{source}$ and a random noise, $\epsilon_{target}$, used to generate the target image. However, when editing an existing image, $x_{source}$, we do not have access to its original noise. A common approach would be to use an inversion method to recover the original noise, $\epsilon_{source}$, that generated $X_{source}$. However, in our experiments, popular inversion methods, such as DDIM inversion (Song et al., 2020), do not adequately reconstruct the image using FLUX. We propose a simple solution: instead of recovering the original noise $\epsilon_{source}$, we sample a random noise $\epsilon$. At each denoising step $t$, we produce a noisy source latent, $X_{source}^t = (1 - \sigma_t)X_{source} + \sigma_t\epsilon$. We then apply our method as usual, using the input batch at timestep $t$, $[X_{source}^t, X_{target}^t]$, where the target image draws information from the source image. This simple technique ensures perfect reconstruction of the source image, since $\sigma_0 = 0$ and therefore $X_{source}^0 = X_{source}$.

Our method, applicable to both generated and real images, can be extended for step-by-step generation. Users can start with an initial image from a textual prompt and iteratively modify it with additional prompts, progressively adding elements or changes to the scene. Examples of step-by-step generation are shown in fig. 1 and fig. 10.

|  | **Emu Edit** | | | | *Additing* **Benchmark** | | | |
|---|---|---|---|---|---|---|---|---|
| Method | $CLIP_{dir}$ | $CLIP_{out}$ | $CLIP_{im}$ | Inc. | $CLIP_{dir}$ | $CLIP_{out}$ | $CLIP_{im}$ | Inc. |
| InstructPix2Pix | 0.074 | 0.312 | 0.929 | 34% | 0.074 | 0.244 | 0.943 | 55% |
| Erasedraw | 0.088 | 0.313 | 0.941 | 65% | 0.117 | 0.248 | 0.958 | 76% |
| Magicbrush | 0.091 | 0.313 | 0.927 | 66% | 0.114 | 0.250 | 0.925 | 86% |
| Paint by Inpaint | 0.071 | 0.316 | **0.955** | 58% | 0.079 | 0.246 | 0.954 | 68% |
| InfEdit | 0.051 | 0.321 | 0.944 | 53% | 0.098 | 0.250 | 0.952 | 54% |
| MasaCtrl | 0.018 | 0.310 | 0.890 | 37% | 0.088 | 0.257 | 0.890 | 66% |
| SDEdit | — | — | — | — | 0.091 | 0.248 | 0.955 | 60% |
| Prompt2Prompt | — | — | — | — | 0.170 | **0.280** | 0.850 | **97%** |
| **Ours** | **0.101** | **0.322** | 0.929 | **81%** | **0.200** | 0.261 | **0.968** | 93% |

Table 2: CLIP and Inclusion metric results for EmuEdit and *Additing* Benchmark.

## 4 EXPERIMENTS

**Evaluation Baselines**  We compare our method with two classes of baselines: **(1)** Training-Free methods that leverage the existing capabilities of text-to-image models: Prompt-to-Prompt (Hertz et al., 2022), a method which injects the attention map of the source image into the target image to preserve its structure, and SDEdit (Meng et al., 2022), a method that adds partial noise to an existing image and then denoises it. Both methods were re-implemented on the FLUX.1-dev model for fair comparison. **(2)** Pretrained Instruction following models, specifically trained to edit and

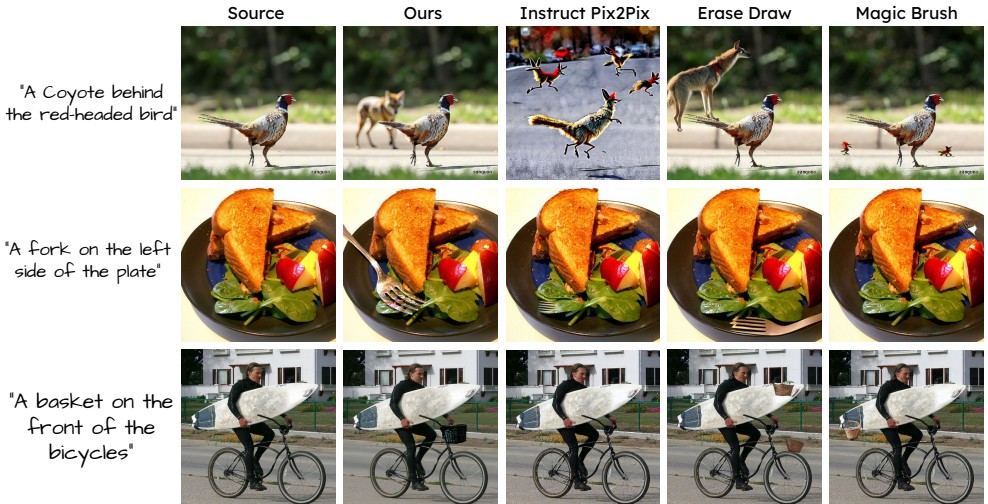

Figure 5: Qualitative Results from the Emu-Edit Benchmark. Unlike other methods, which fail to place the object in a plausible location, our method successfully achieves realistic object insertion.

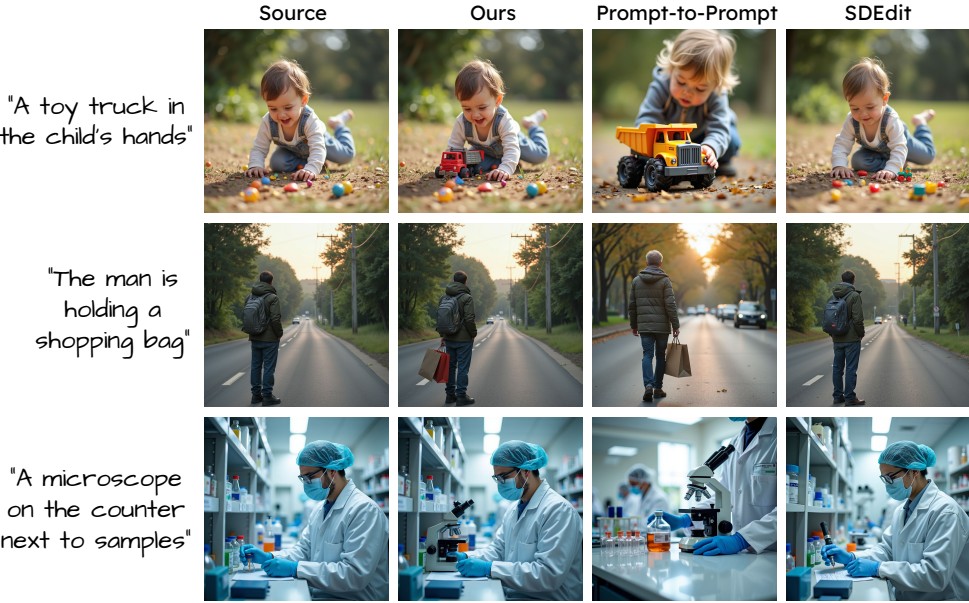

Figure 6: Qualitative Results from the *Additing* Benchmark. While Prompt-to-Prompt fails to align with the source image, and SDEdit fails to align with the prompt, our method offers *Additing* that adheres to both prompt and source image.

add objects to existing images: InstructPix2Pix (Brooks et al., 2023) an instruction following model trained on a large scale of synthetic instruction data, Magicbrush (Zhang et al., 2023) a version of InstructPix2Pix fine-tuned on manually annotated editing dataset, and Erasedraw (Canberk et al., 2024) a model trained on large dataset constructed using an inpainting model. *Add-it* implementation details can be found in appendix A.1.

**Metrics**    We evaluate the results of our method and the baselines using automatic metrics and human evaluations for each source and target image-caption pair. *Automatic Metrics*: we start by adopting the CLIP (Radford et al., 2021) based metrics proposed in Emu-Edit (Sheynin et al., 2023): (i) **CLIP**$_{dir}$ (Gal et al., 2022) measures the agreement between change in captions and the change in images. (ii) **CLIP**$_{img}$ measures similarity between source and target images. (iii) **CLIP**$_{out}$ measures the target image and caption similarity. We propose two additional metrics: (iv) **Inclusion** measures the portions of cases the object was added to the image, evaluated automatically using the open-

vocabulary detection model Grounding-DINO (Liu et al., 2023). (v) **Affordance** measures whether the object was added to a plausible location, utilizing Grounding-DINO and a manually annotated set of possible locations. *Human Evaluations*: we ask human raters to pick the best *Additing* output when faced with a source image, instruction and images generated by our method and a competing baseline. Further details in appendix A.8.

## 4.1    EVALUATION RESULTS

**Emu-Edit Benchmark**    Following EraseDraw (Canberk et al., 2024) we evaluate our method on a subset of EmuEdit's (Sheynin et al., 2023) validation set with the task class of "Add", designed for insertion instructions. The benchmark consists of sets of images and prompts before and after an edit, and the corresponding instruction. Table 2 shows our model outperforms all previous approaches in the $CLIP_{dir}$, $CLIP_{out}$ and the Inclusion metrics. In the $CLIP_{im}$ metric, which indicates how close the edited image is to the source image, we are second only to Erasedraw. This result is not surprising given that in 35% of the cases Erasedraw did not add an object to the image (indicated by the Inclusion metric), artificially boosting the image similarity score. Due to the limitations of automatic metrics, especially in assessing the naturalness of edits, we conducted a head-to-head evaluation with human raters against each baseline, as shown in fig. 3. Our method's outputs were preferred in  80% of cases.  Finally, we present a qualitative comparison to other methods using images from the EmuEdit benchmark in fig. 5. Previous methods often produce artifacts, unnatural object placements, or fail to modify the image. In contrast, our method generates high-quality outputs that consider the context of the source image.

*Additing* **Benchmark**    To evaluate our method against both pre-trained models and zero-shot methods, which tend to perform better on generated images, we created a benchmark for the *Additing* task. We asked ChatGPT to generate 200 sets of source and target prompts along with *Additing* instructions. Using Flux, we generated images and filtered 100 sets where the instructions were viable. We report all results in Table 2. Our method outperforms all baselines on the $CLIP_{dir}$ and $CLIP_{im}$ metrics. Although Prompt-to-Prompt slightly surpasses us on $CLIPout$ and Inclusion, it does so by heavily altering the source image, as shown by its low $CLIPim$ score. As in the EmuEdit Benchmark, we asked human raters to compare our method against the zero-shot baseline. Our method was preferred in 90% of cases against Prompt2Prompt and 83% against SDEdit fig. 4. Finally, fig. 6 shows a comparison on the *Additing* Benchmark, where other methods struggle to balance object addition, background preservation, and context, while ours produces natural, appealing outputs.

*Additing* **Affordance Benchmark**    Throughout our experiments we observed that the major shortcoming of existing methods is incorrect affordance, namely, objects are added at implausible locations (see the basket in  fig. 5). To automatically quantify affordance, we constructed an affordance benchmark.  It contains 200 images and prompts, with manually annotated bounding-boxes indicating the plausible locations to add objects in each image.  Dataset construction and evaluation protocol details are available in appendix A.6. We present the results of all methods in  table 1. As expected, previous methods perform poorly, with low affordance scores, particularly trained models like InstructPix2Pix, which scored as low as 0.276. In contrast, *Add-it* scores nearly double that of the best-performing method, demonstrating its ability to balance information from the source image and target prompt. We present additional results of our method, including comparisons on the MagicBrush dataset and image quality metrics, in appendix A.2 and appendix A.3.

## 5    ANALYSIS

In this section, we analyze the attention distribution in the *MM-DiT* block and the key components of our method to better justify our design choices. In appendix A.4 we analyze the role of positional encoding in the extended-attention mechanism.

*MM-DiT* **Attention Distribution**    First, we analyze the different attention components in the extended *MM-DiT* blocks. Recall that in the extended-attention mechanism described in section 3.2 there are three token sets: the source image $X_{source}$, the target image $X_{target}$ and the prompt $P_{target}$. In our experiments, we notice that simply applying the extended attention mechanism results in the target image closely following the appearance of the source image while neglecting the

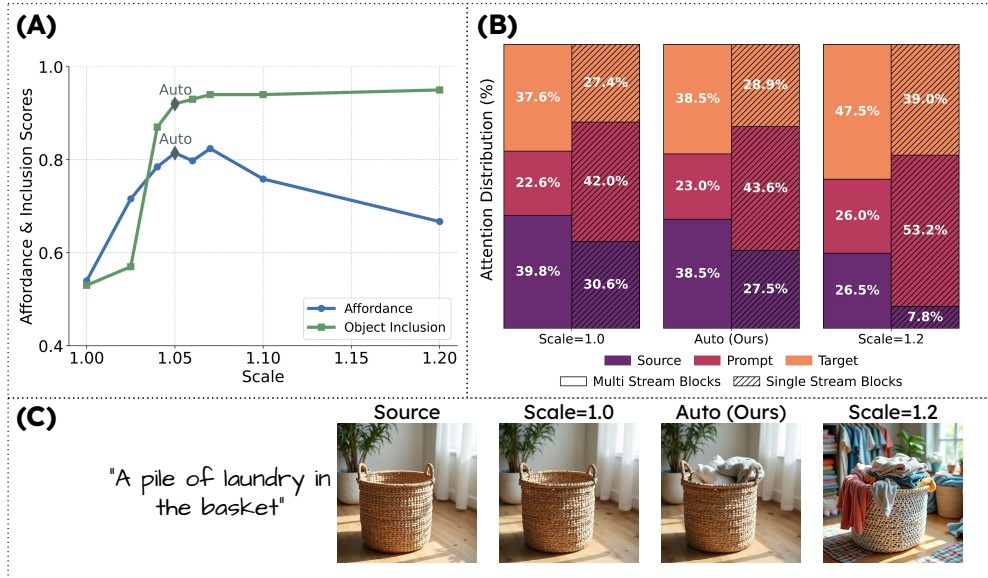

Figure 7: **(A)** Affordance and Object Inclusion scores across weight scale values, with our automatic weight scale achieving a good balance between the two. **(B)** Visualization of the prompt token attention spread across different sources, model blocks, and weight scales, averaged over multiple examples from a small validation set. **(C)** A representative example demonstrating the effect of varying target weight scales.

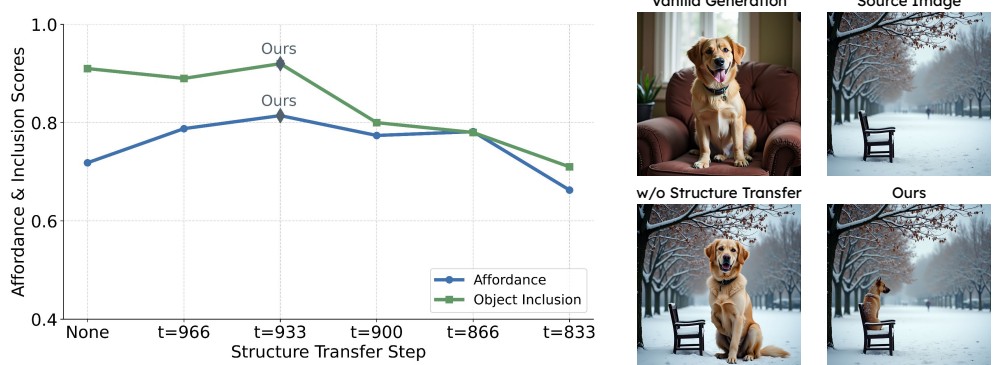

Figure 8: Ablation over various steps for applying the Structure Transfer mechanism. Applying it too early misaligns the generated images with the source image's structure while applying it too late causes the output image to neglect the object. Our chosen step strikes a balance between both.

prompt - meaning no object is added to the image. We attribute this problem to the way the attention is distributed across the three sets of tokens. In particular, we find empirically that the target prompt's attention $A_p \propto \exp(Q_p \cdot [K_{source}, K_p, K_{target}])$ serves as an effective proxy for balancing the three sources of attention. A simple way to control the attention distribution is by introducing scale factor $\gamma_p, \gamma_{target}$ so that $A_p \propto \exp(Q_p \cdot [K_{source}, \gamma_p \cdot K_p, \gamma_{target} \cdot K_{target}])$. In practice, we find that using $\gamma = \gamma_p = \gamma_{target}$ is adequate. In fig. 7 (B) we visualize the prompt attention $A_p$ spread across the three token sets. In the standard extended-attention case ($\gamma = 1.0$), the source image tokens (purple) receive more attention than the target image tokens (orange), preventing the generated image from incorporating the added object. On the other hand when scaling up too much ($\gamma = 1.2$), the target image tokens overwhelm the source image token, causing the output image to stray away from the source image structure. Finally, when the scaling value balances the attention between $X_{source}$ and $X_{target}$ ($\gamma = \text{Auto}$), the output image successfully incorporates the added object, while preserving the target image structure and taking into account its context when placing the object. These observations are qualitatively shown in fig. 7 (C) and are also reflected in fig. 7 (A), where the scale that balances the attention offers a good balance between affordance and Object Inclusion.

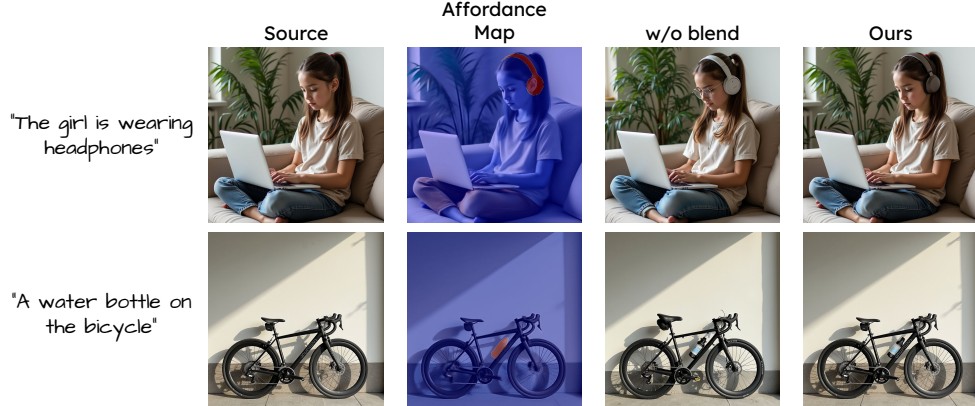

Figure 9: Images generated by *Add-it* with and without the latent blending step, along with the resulting affordance map. The latent blending block helps align fine details from the source image, such as removing the girl's glasses or adjusting the shadows of the bicycles.

**Ablation Study**   Next, we evaluate the impact of different components of our method. First, we demonstrate the effect of the weight scale, $\gamma$. In fig. 7 (A) we present a graph showing affordance and Object Inclusion as functions of different weight scales. As the weight scale increases, the added object tends to appear more frequently in the image. However, beyond a certain threshold, the affordance score drops. This decline occurs when the target image ignores the structure of the source image, generating objects in unnatural locations, as illustrated in fig. 7 (C). Next, we explore the effect of latent blending. In fig. 9 we show output images with and without the latent blending step, along with the affordance map automatically extracted by our method. Notice how the blending step aligns the fine details of the source image without introducing artifacts. An ablation of the localization mask construction is provided in appendix A.5. Finally, we examine the structure transfer component. In fig. 8 we illustrate the effect of applying the structure transfer step at different denoising process stages. When structure transfer is applied too early, the affordance score is low, meaning the target image does not adhere to the structure of the source image. On the other hand, applying it later in the process results in a lower object inclusion metric, indicating that the target image neglects the object. Ultimately, when the structure transfer is applied at $t = 933$, we achieve a balance between object inclusion and affordance. A qualitative example is provided in fig. 8.

## 6   LIMITATIONS

*Add-it* shows strong performance across various benchmarks, but it has some limitations. Since the method relies on pretrained diffusion models, it may inherit biases its biases, potentially affecting object placement in unfamiliar or highly complex scenes, as well as introducing biases such as gender bias (fig. 15). Additionally, because our method uses target prompts rather than explicit instructions, users may need to construct more detailed prompts to achieve the same edit. For instance, with an image of a dog, the prompt "A dog" won't add another dog to the scene. Instead, the user would need to provide a more specific prompt, such as "A second dog beside the dog in the middle."

## 7   CONCLUSION

We introduced *Add-it*, a training-free method for adding objects to images using simple text prompts. We analyzed the attention distribution in *MM-DiT* blocks and introduced novel mechanisms such as weighted extended-attention and Subject-Guided Latent Blending. Additionally, we addressed a critical gap in evaluation by creating the "Additing Affordance Benchmark," which allows for an accurate assessment of object placement plausibility in image *Additing* methods. *Add-it* consistently outperforms previous approaches, improving affordance from 47% to 83% and achieving state-of-the-art results on both real and generated image benchmarks. Our work demonstrates that leveraging the knowledge in pretrained diffusion models is a promising direction for tackling challenging tasks like image *Additing*. As diffusion models continue to evolve, methods like *Add-it* have the potential to drive further advancements in semantic image editing and related applications.

## ETHICS STATEMENT

In this work, we acknowledge the ethical considerations associated with image editing technologies. While our method enables advanced object insertion capabilities, it also has the potential for misuse, such as creating misleading or harmful visual content. We strongly encourage the responsible and ethical use of this technology, emphasizing transparency and consent in its applications. Additionally, biases present in pretrained models may affect generated outputs, and we recommend further research to mitigate such issues in future work. Human evaluations were conducted with informed consent.

## REPRODUCIBILITY STATEMENT

All necessary information to reproduce *Add-it* is provided in section 3 and appendix A.1. We provide the proposed "*Additing* Benchmark" and "*Additing* Affordance Benchmark" in the supplementary material of our submission.

We will open-source all the code upon publication of the paper.

## ACKNOWLEDGMENTS

We thank Assaf Shocher, Lior Hirsch and Omri Kaduri for useful discussions and for providing feedback on an earlier version of this manuscript. This work was completed as part of the first author's PhD thesis at Tel-Aviv University.

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

## A APPENDIX

### A.1 IMPLEMENTATION DETAILS

***Add-it*** When evaluating *Add-it*, we use $t_{struct} = 933$ for generated images and $t_{struct} = 867$ for real images and $t_{blend} = 500$. For the scaling factor $\gamma$, we use the root-finding solver described in section 3.2 on a set of validation images and set $\gamma$ to 1.05, as it is close to the average result and performs well in practice. We generate the images with 30 denoising steps, building upon the diffusers implementation of the FLUX.1-dev model. We apply the extended attention mechanism until step $t = 670$ in the multi-stream blocks, and step $t = 340$ for the single-stream blocks.

**Computation Time** To compare the computational efficiency of *Add-it* with the original FLUX model, we generated 200 images using each method and reported the average generation time along with the standard error of the mean. The base FLUX model requires 6.4±0.13 seconds to generate a single image, while *Add-it* takes 7.23±0.05 seconds per image—an increase of just under one second. This slight increase is primarily due to caching attention maps and the additional computations required for the extended attention mechanism. SAM2 contributes only minimally to this difference, with an average inference time of 0.04 seconds.

**Latent Blending Localization** To extract a refined object mask as part of the Subject Guided Latent Blending component, we begin by extracting subject attention maps. Empirically, we find that the best-performing layers for this task are: `["transformer_blocks.13","transformer_blocks.14", "transformer_blocks.18", "single_transformer_blocks.23", "single_transformer_blocks.33"]`. To refine the mask from these attention maps, we need to identify points to use as prompts for SAM-2. To extract points from the attention map, we first select the point with the highest attention value. Then, we exclude the area around the chosen point and select the next highest point. This process is repeated until we either identify 4 points or the current maximal point value falls below $0.35 \cdot p_{max}$, where $p_{max}$ is the initial maximum attention value. Finally, we feed the points to the SAM-2 model to end up with a refined object mask.

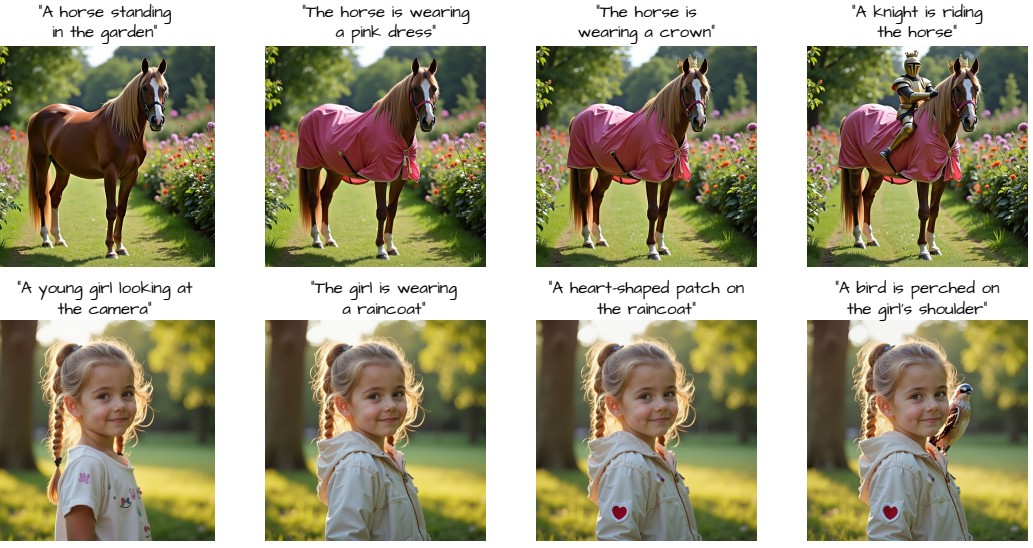

Figure 10: **Step-by-Step Generation:** *Add-it* can generate images incrementally, allowing it to better adapt to user preferences at each step.

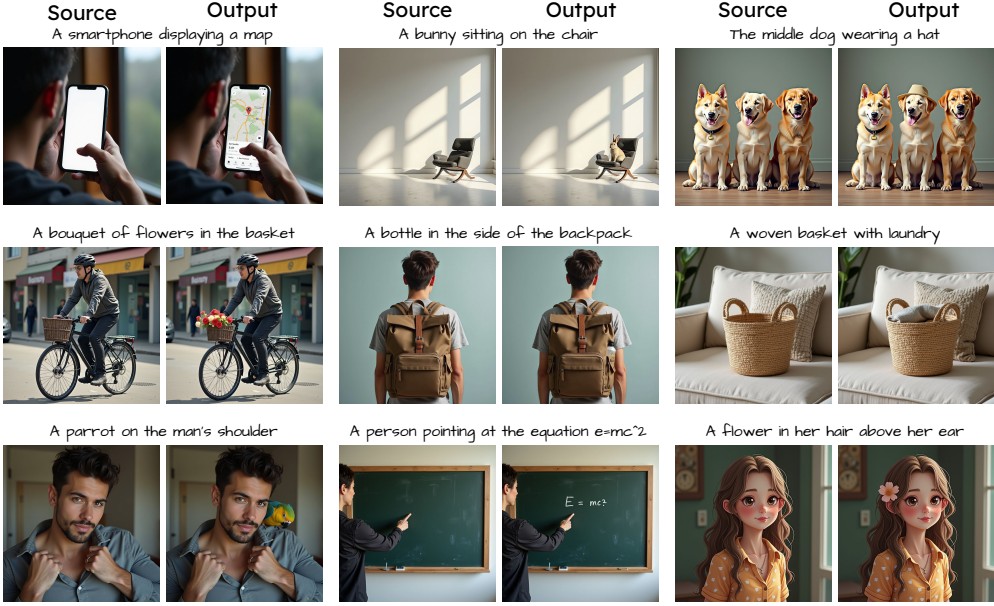

Figure 11: Qualitative results of our method on the *Additing* Affordance Benchmark show that our method successfully adds objects naturally and in plausible locations.

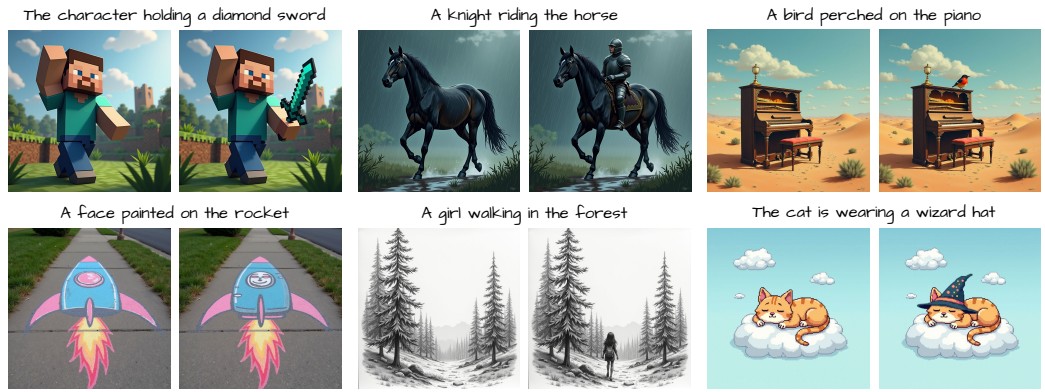

Figure 12: Our method can operate on non-photorealistic images.

## A.2 ADDITIONAL RESULTS

In fig. 10 we present step-by-step outputs generated with *Add-it*. Notice that the scene remains unchanged, while each prompt adds an additional "layer" to the final image, resulting in a more complex scene.

In fig. 11 we show additional results from the *Additing* Affordance benchmark. In each case, the object must be added to a specific location in the source image. Across all examples, *Add-it* successfully places the object in a plausible location, preserving the natural appearance of the image.

In fig. 12 we demonstrate that *Add-it* can operate on non-photorealistic source images, such as paintings and pixel art. Since our method requires no tuning, we preserve all the generation capabilities of the base model.

In fig. 13 we show various generation results produced by our model, each originating from a different initial noise. Our method preserves the diversity of the base model, enabling users to generate multiple variations of the added object until they find the desired one.

In fig. 14 we demonstrate *Add-it*'s ability to edit existing objects in an image. Although our method was originally designed for object *Additing*, it can also perform edits such as changing a person's hair color, eye color, or his shirt.

In fig. 15 we highlight that biases in the pre-trained diffusion model can also manifest in *Add-it*. In this figure, we prompt our method to add a doctor to an empty hospital room without specifying gender or any additional details. We display multiple random generations from different noise initializations and observe that, in all cases, the generated doctor is male. We attribute this bias to underlying biases in FLUX, which our method inherits.

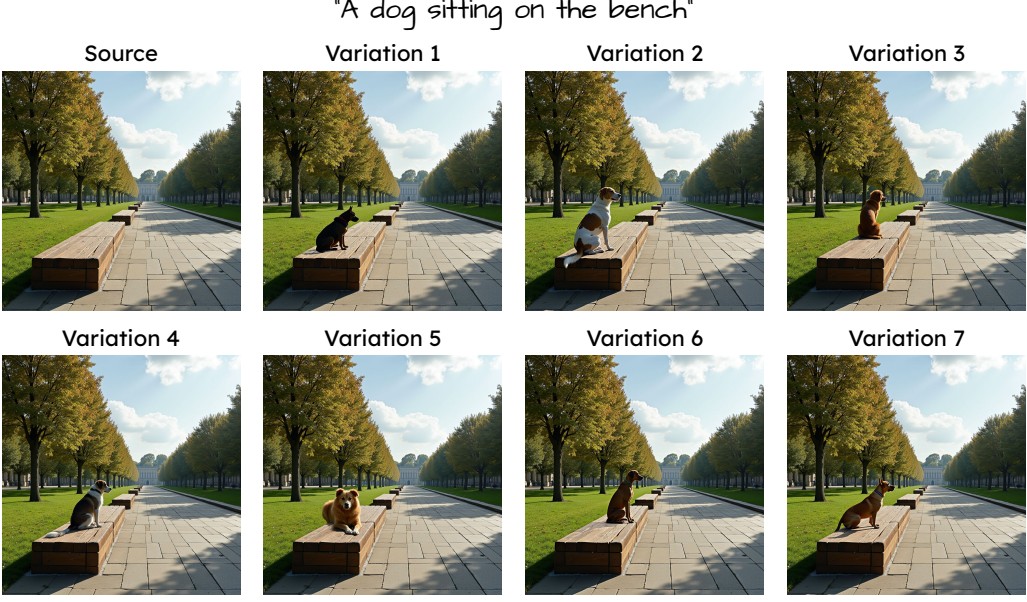

Figure 13: Our method generates different outputs when given different starting noises. All the outputs remain plausible.

### A.3 Additional Comparisons

To further evaluate *Add-it*, we compare it against three additional baselines: MasaCtrl (Cao et al., 2023), a training-free editing method that conditions on a source image using a mutual self-attention mechanism; InfEDIT (Xu et al., 2024), which leverages Latent Consistent Models for training- and inversion-free image editing; and Paint by Inpaint (Wasserman et al., 2024), a model trained on a large inpainting dataset to add objects to images. We present the affordance scores of these baselines in table 3, evaluated on the *Additing* Affordance Benchmark. Consistent with previous evaluations, these methods achieve low scores, demonstrating their inability to consistently insert objects into plausible locations in the scene. We further evaluate the three baselines on both the EmuEdit dataset and the *Additing* benchmark, with the results shown in table 4. *Add-it* outperforms all baselines, with the exception of the $CLIP_{im}$ metric on the EmuEdit dataset. Notably, the inclusion metric for these baselines lags significantly behind our method, highlighting a common failure case where these methods are unable to successfully add the new object to the scene.

We expanded our evaluation by comparing *Add-it* to existing baselines on the test split of the MagicBrush Benchmark, filtering for examples with insertion instructions only. In table 5 we present a quantitative comparison of *Add-it* against other baselines using CLIP and Inclusion metrics. Notably, our method outperforms all baselines, including the MagicBrush model itself. In fig. 16 provides a qualitative comparison, showcasing examples where *Add-it* successfully adds objects like a dog and a squirrel in plausible locations, seamlessly integrating them into the scene. In contrast, other methods struggle with both coherent object generation and proper placement.

Finally, we include two widely used metrics to assess image quality and diversity: KID(Bińkowski et al., 2018), which measures how closely generated images match the distribution of real images

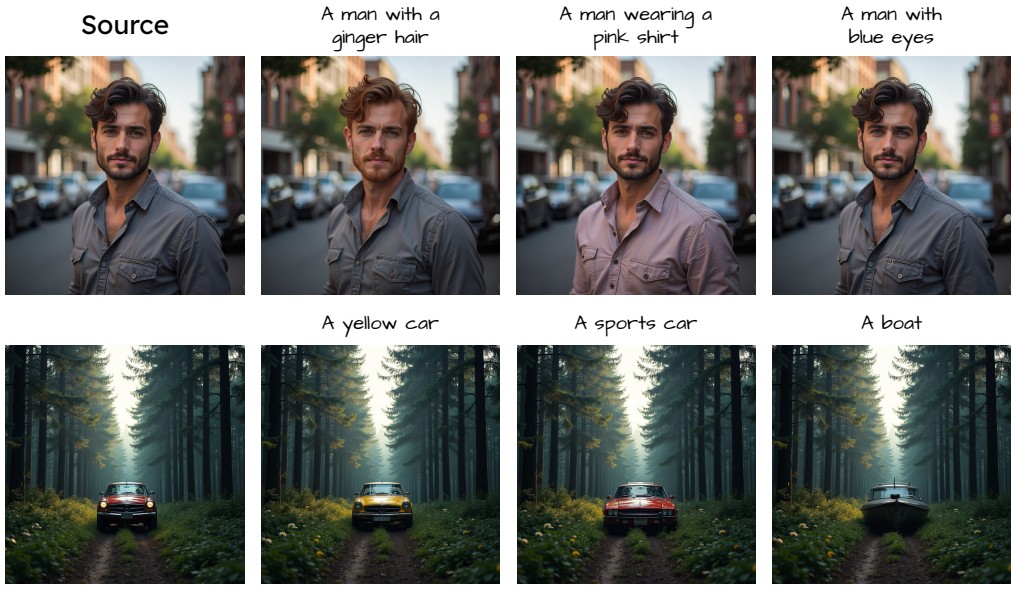

Figure 14: *Add-it* can perform editing of existing objects in the images, such as changing the man's hair color, or switching the car for a boat.

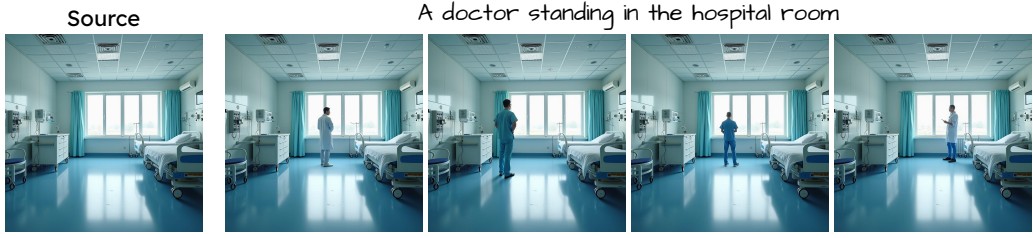

Figure 15: *Add-it* can suffer from bias of the underlying diffusion model, such as generating mostly male doctors.

(lower is better), and Inception Score (IS)(Salimans et al., 2016), which evaluates both image quality and diversity based on a pre-trained inception model (higher is better). We chose KID over FID as it performs more reliably on smaller evaluation sets. While these metrics are traditionally used for image generation, our focus is on image *Adding*. To adapt them for our task, we use an off-the-shelf open vocabulary detection model to identify regions containing added objects across the dataset. We then crop these regions to create a dataset of object crops and evaluate KID and IS by comparing this dataset against the source images before editing. This approach allows us to assess the quality and diversity of the inserted objects directly. Our method, along with SDEdit and Prompt2Prompt, achieves the lowest KID scores, indicating high quality of the added objects. Additionally, *Add-it* and Prompt2Prompt attain the highest IS scores, demonstrating that the inserted objects are also diverse.

|  | InfEdit | MasaCtrl | Paint By Inpaint | **Ours** |
|---|---|---|---|---|
| Affordance | 0.366 | 0.203 | 0.311 | **0.828** |

Table 3: Comparison of methods based on Affordance score for the *Adding* Affordance Benchmark.

| Method | Emu Edit | | | | *Adding* Benchmark | | | |
|---|---|---|---|---|---|---|---|---|
| | $\text{CLIP}_{dir}$ | $\text{CLIP}_{out}$ | $\text{CLIP}_{im}$ | Inc. | $\text{CLIP}_{dir}$ | $\text{CLIP}_{out}$ | $\text{CLIP}_{im}$ | Inc. |
| InfEdit | 0.051 | 0.321 | 0.944 | 53% | 0.098 | 0.250 | 0.952 | 54% |
| MasaCtrl | 0.018 | 0.310 | 0.890 | 37% | 0.088 | 0.257 | 0.890 | 66% |
| Paint by Inpaint | 0.071 | 0.316 | **0.955** | 58% | 0.079 | 0.246 | 0.954 | 68% |
| **Ours** | **0.101** | **0.322** | 0.929 | **81%** | **0.200** | **0.261** | **0.968** | **93%** |

Table 4: CLIP and Inclusion metric results for EmuEdit and *Adding* Benchmark.

| Method | $\text{CLIP}_{dir}$ | $\text{CLIP}_{out}$ | $\text{CLIP}_{im}$ | Inc. |
|---|---|---|---|---|
| InstructPix2Pix | 0.077 | 0.297 | 0.917 | 45% |
| EraseDraw | 0.117 | 0.301 | 0.934 | 58% |
| MagicBrush | **0.144** | 0.303 | 0.905 | 72% |
| Paint by Inpaint | 0.098 | 0.300 | 0.928 | 62% |
| **Ours** | 0.124 | **0.307** | **0.937** | **85%** |

Table 5: CLIP and Inclusion metric results for the MagicBrush Benchmark.

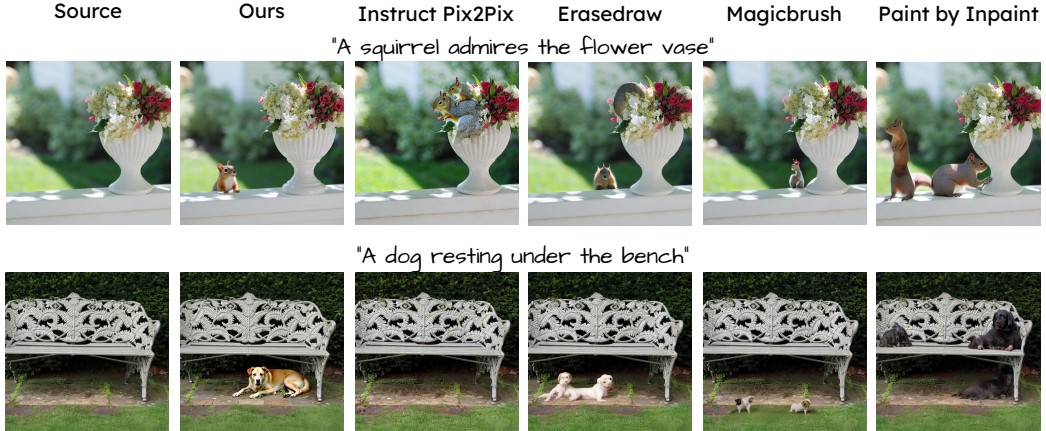

Figure 16: Qualitative results from the MagicBrush benchmark show that our method generates high-quality objects placed correctly within the scene, whereas other methods struggle with both object quality and placement.

| Method | KID ↓ | IS ↑ |
|---|---|---|
| InstructPix2Pix | 0.034 | 4.173 |
| EraseDraw | 0.027 | 4.473 |
| MagicBrush | 0.040 | 4.243 |
| Paint by Inpaint | 0.030 | 3.762 |
| SDEdit | 0.016 | 2.783 |
| Prompt2Prompt | 0.014 | 4.901 |
| **Ours** | 0.015 | 4.949 |

Table 6: KID and Inception Score metrics used to assess the quality and diversity of edits by *Add-it* and other baselines.

## A.4 THE ROLE OF POSITIONAL ENCODING

Here, we examine the significance of positional encodings in the extended attention mechanism. fig. 17 demonstrates their role through a simple experiment: we applied our method to a source image where the positional encoding vectors were shifted down and to the right. This misalignment resulted in a mismatch between the positional encoding of the child's head in the source and target

| Source Image | w/ PE Shift | w/o PE Shift |
|:---:|:---:|:---:|
| 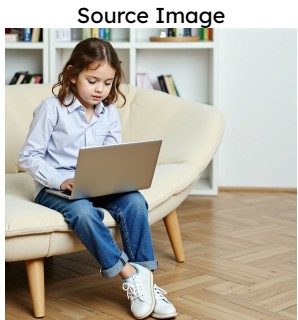 | 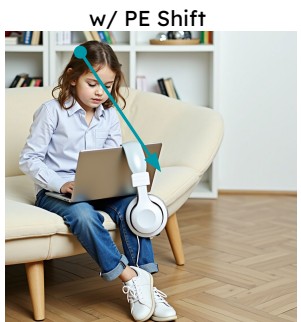 | 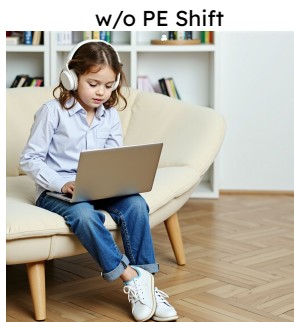 |

Figure 17: Positional Encoding Analysis: shifting the positional encoding of the source image results in a corresponding shift in the object's location in the generated image.

images. Consequently, instead of generating headsets at the actual position of the child's head, the model produced them in the area corresponding to the "shifted head" position. This outcome demonstrates that the model heavily relies on positional information to transfer features between the source and target images. Despite the target image containing "laptop" features instead of "head" features at the relevant location, the model chose to place the headphones there. This decision was based on the area having the same positional encoding as the "head area" in the source image, rather than on the actual content of the target image at that location. We believe further research on the role of positional encoding vectors is an interesting direction for future work in the context of DiT models.

### A.5 LATENT BLENDING MASK CONSTRUCTION ABLATION

Here, we examine different methodologies for generating the object mask used as input to the Latent Blending step. We qualitatively display these options in fig. 18. First, we compare the results using the coarse attention mask extracted from the attention layers, as described in appendix A.1.Notably, the attention maps tend to capture the main parts of the person but often exclude details such as legs, resulting in incomplete and messy masks. Next, we consider the mask generated by SAM2 when prompted with the coarse mask as input. In many cases, the coarse mask proves to be an inadequate prompt for SAM2, leading to refined masks that still fail to capture the entire object. Finally, we present our chosen methodology, discussed in appendix A.1, where we extract key points from the attention mask and use them as input to SAM2. This approach produces a refined mask that captures the entire object, resulting in a seamless integration of the person into the scene. Additionally, table 7 shows the affordance scores for each methodology, demonstrating that our proposed point-based algorithm yields the best results.

| | Coarse Attention Mask | Mask Prompt SAM | **Ours** |
|:---:|:---:|:---:|:---:|
| Affordance | 0.809 | 0.77 | **0.828** |

Table 7: Ablation of blending mask construction methods based on Affordance score for the *Additing* Affordance Benchmark.

### A.6 *Additing* AFFORDANCE BENCHMARK

**Affordance in Image *Additing*** In section 1, we highlight a significant gap in current object insertion evaluation protocols: the lack of a mechanism to assess whether an object was added to a plausible location. Our experiments reveal that existing methods often struggle to find the "right" spot for object placement, which we refer to as affordance. To address this, we introduced the "Adding Affordance Benchmark". The guiding principle behind the benchmark is the need for an automated method to evaluate whether an image editing method has inserted an object in a plausible location within the scene. Since developing an automated approach to assess correct object placement is itself an unsolved challenge, we created a dataset consisting of images, insertion instructions, and carefully labeled regions indicating plausible insertion areas for each instruction. This allows us to

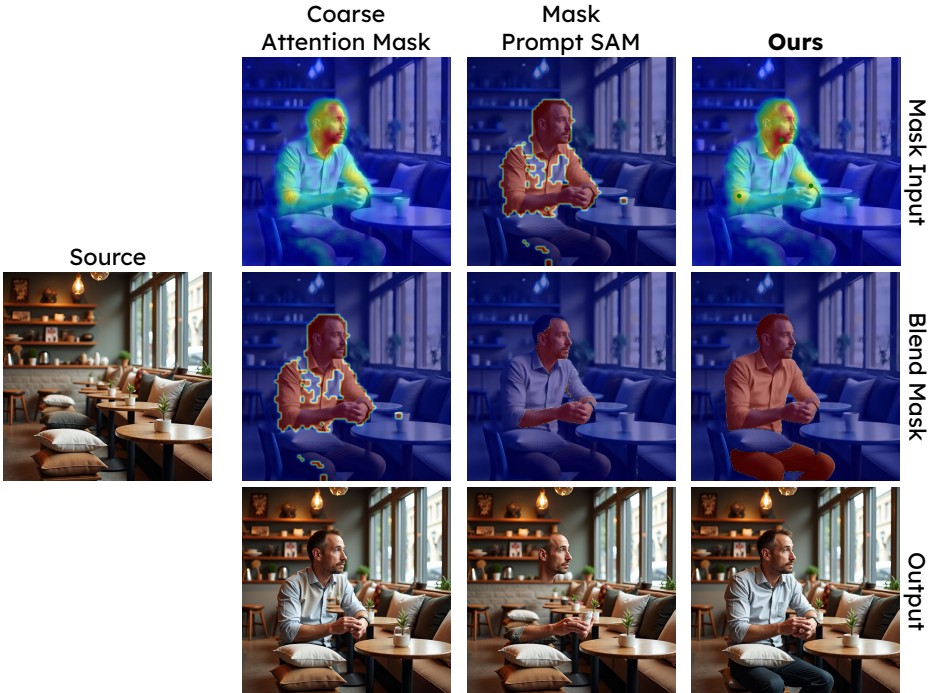

Figure 18: Qualitative ablation of blending mask construction: While the coarse attention mask and the mask-prompted SAM mask may not perfectly segment the added object, our point-based algorithm typically generates precise masks, leading to a seamless integration of the object into the scene.

use an off-the-shelf open-vocabulary detector (as described below) to determine whether an added object's bounding box lies within the manually labeled region, thereby indicating whether it was placed in a plausible location. As shown in section 4, the *Additing* Affordance Benchmark reveals that all existing image addition methods struggle with this requirement, highlighting a critical gap in the field that needs to be addressed to develop effective object insertion methods.

**Dataset Construction**   Here, we provide the details for constructing the *Additing* Affordance Benchmark dataset. First, we used ChatGPT-4 to generate a dataset of tuples, each consisting of a source prompt and a target prompt, representing an image before and after object insertion, along with an instruction for the transition and a subject token representing the object to be added. The exact prompt is shown in fig. 20. Next, we used FLUX.1-dev to generate the source images from the source prompts in each tuple. We manually filtered out images where the object had no plausible location or too many possible locations, resulting in a dataset of 200 images. Finally, we manually annotated bounding boxes for each image, marking the plausible locations where the object could be added, as shown in fig. 19.

**Evaluation protocol:**   Given a set of an *Additing* model output images, we use Grounding-DINO to detect the area where new objects were added and set the affordance score of a single image to be the fraction of added object that at least 0.5 of their area falls inside the GT box.

### A.7   PROMPTING WITH *Add-it*

In contrast to instruction-based editing methods, *Add-it* operates on target prompts that describe the edited image, including the added object. Example prompts appear in the qualitative figures throughout the paper. In fig. 21 we demonstrate that an LLM, such as ChatGPT, can easily convert between these representations. Moreover, our EmuEdit evaluation is conducted using automatically converted captions with the exact same prompt - showing its effectiveness.

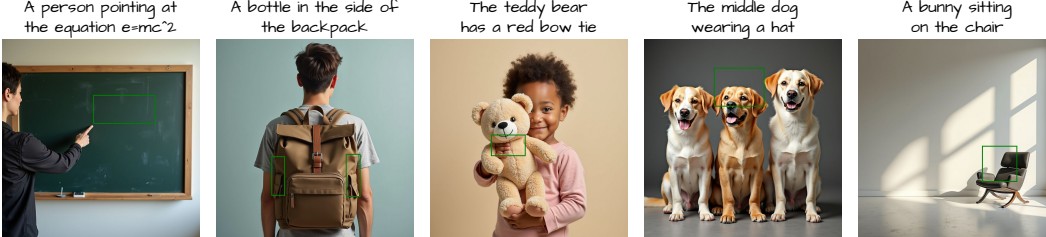

Figure 19: Visual examples from the *Additing* Affordance Benchmark. Each image is annotated with bounding boxes highlighting the plausible areas where the object can be added.

```
Please generate a JSON list of 300 sets. Each set consists
of: an index, a source prompt, instruction, a target prompt,
and a subject token.
The source prompt describes a source image.
The target prompt describes the source image after an object
has been added to it.
The instruction is a description of what needs to be changed
to go from the source to the target prompt.
The subject token is the noun that refers to the added
object, a single word that appears in the target prompt.
Here are is an example:
{
    "src_prompt": "A person sitting on a chair",
    "tgt_prompt": "A scarf wrapped around their neck",
    "subject_token": "scarf",
    "instruction": "Wrap a scarf around the person's neck."
}
Only generate examples where there is clearly
only one possible place for the object to be added, so it
can be tagged correctly.
Write it as a JSON list yourself.
Please DO NOT include negative examples in your prompts,
such as "a man wearing no hat" in the source prompt.
DO NOT write code; Return only the JSON list.
```

Figure 20: The prompt provided to ChatGPT in order to generate the Affordance Benchmark.

## A.8 USER STUDY DETAILS

We evaluate the models through an Amazon Mechanical Turk user study using a two-alternative forced choice protocol. In the study, raters saw an instruction, a source image, and two edited images, each produced by a different approach. They chose the edit that best followed the instruction, taking into account: image quality and realism, instruction following and preservation of the source image. For the evaluation, each head-to-head example was rated by two raters. In fig. 22 we show an example of a single trial a rater has seen.

```
Given an input caption and an editing instruction, generate
an output caption that reflects how an input image,
corresponding to the input caption, would appear after
being edited according to the instruction.

Here are few example:
input_caption: A hotel bed ready for use
instruction: Add a suitcase to the bed
output_caption: A hotel bed with a suitcase on it

input_caption: A table with two plates of breakfast food
instruction: Add a fork to the table between the plates
output_caption: A table with two plates of breakfast food
with a fork between them.
```

Figure 21: The prompt provided to ChatGPT in order to generate output prompts from instruction-based prompts appearing in benchmark such as EmuEdit.

Select the edited image that best follows the insertion instruction below.
Focus on how well the object was inserted while maintaining quality the appearance of the source image.

## Instructions

- **Read these instructions carefully.**
- You will see **two** edited images based on a **source image** and an **insertion instruction**.
- Pick the image that best follows the instruction.
- **Focus on these factors:**
  - **Image quality and realism:** The edit must look high-quality and natural. It should blend in perfectly with no visible flaws or distortions.
  - **Instruction accuracy:** The image must match the object addition exactly as described in the instruction.
  - **Preservation of the source image:** The rest of the image should have only necessary changes.
- **Choose only one image.**

### Instruction: Add a football jersey to the skateboarder

**Source Image:**       **Edit 1**

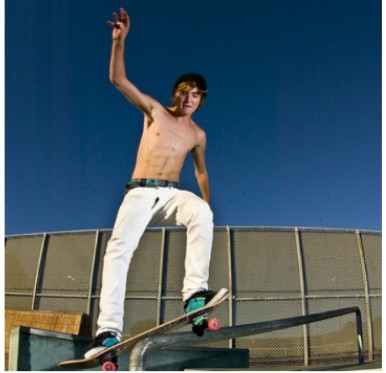

**Source Image:**       **Edit 2**

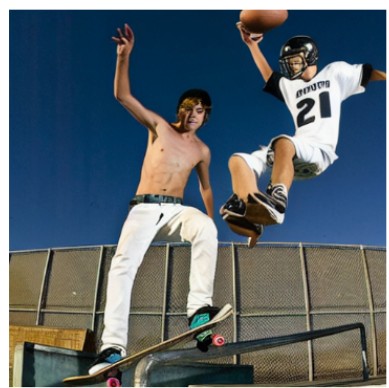

○ Choose Edit 1    ○ Choose Edit 2

[ Submit ]

Figure 22: One trial of the *Additing* user study.

