# OpenReview forum: "Add-it: Training-Free Object Insertion in Images With Pretrained Diffusion Models"
_ICLR.cc/2025/Conference — ICLR 2025 Poster_

### Official Review · Reviewer_9iYS · 2024-10-18

**Soundness:** 2
**Presentation:** 3
**Contribution:** 2
**Rating:** 6
**Confidence:** 4

**Summary:**

The paper proposes a tuning-free method for adding objects into images based on text instructions. They introduce Add-it, a training-free approach that extends the attention mechanisms of diffusion models to incorporate information from three key sources: the scene image, the text prompt, and the generated image itself. The experimental results show that it effectively maintains structural consistency and fine details while ensuring natural object placement compared to existing approaches.

**Strengths:**

1. The paper proposes a training-free method that achieves state-of-the-art results in object insertion.
2. The analysis of MM-DiT Attention Distribution is interesting and introduces an effective mechanism to control their contribution.
3. The paper Introduces a benchmark to assess the plausibility of object insertion.
4. The writing and presentation are mostly clear and easy to follow. The authors provide hyperparameters and detailed settings used in experiments and method implementation, which is helpful for reproducibility.

**Weaknesses:**

The proposed techniques are incremental, lacking sufficient novelty to make a significant contribution to the field. The WEIGHTED EXTENDED SELF-ATTENTION, STRUCTURE TRANSFER, and SUBJECT GUIDED LATENT BLENDING methods are commonly used in Unet-based free-tuning image editing [1,2,3,4,5], and the modifications presented here do not represent a substantial advancement.

- In the WEIGHTED EXTENDED SELF-ATTENTION section, the approach of applying a coefficient to the key is similar to existing techniques in the community, such as scaling text embeddings [1]. This makes the proposed contribution appear incremental rather than innovative.
- In the STRUCTURE TRANSFER section, the proposed technique shows only minor improvements compared to SDEdit [2], without introducing a fundamentally new approach.
- In the SUBJECT GUIDED LATENT BLENDING section, similar blending operations have been previously demonstrated in [3, 4, 5]. Furthermore, using SAM to assist image blending, while effective, is a straightforward extension and lacks the originality needed to stand out.
- The selection of tuning-free editing baselines is also inadequate. Methods such as Masactrl [3], Proximal Guidance [4], and InfEdit [5] provide more sophisticated attention or mask control mechanisms. I recommend including a more comprehensive comparison with these existing baselines to better contextualize the contribution of the proposed method.

Other error:
Incorrect usage of quotation marks on line 237: The word 'choose' is improperly quoted.

[1] https://huggingface.co/docs/diffusers/using-diffusers/weighted_prompts#prompt-weighting

[2] C. Meng, Y. He, Y. Song, J. Song, J. Wu, J.-Y. Zhu, and S. Ermon, "SDEdit: Guided Image Synthesis and Editing with Stochastic Differential Equations," International Conference on Learning Representations (ICLR), 2022.

[3] M. Cao, X. Wang, Z. Qi, Y. Shan, X. Qie, and Y. Zheng, "MasaCtrl: Tuning-Free Mutual Self-Attention Control for Consistent Image Synthesis and Editing," Proceedings of the IEEE/CVF International Conference on Computer Vision (ICCV), 2023, pp. 22560-22570.

[4] L. Han, S. Wen, Q. Chen, Z. Zhang, K. Song, M. Ren, R. Gao, A. Stathopoulos, X. He, Y. Chen, D. Liu, Q. Zhangli, J. Jiang, Z. Xia, A. Srivastava, and D. N. Metaxas, "ProxEdit: Improving Tuning-Free Real Image Editing with Proximal Guidance," Proceedings of the IEEE/CVF Winter Conference on Applications of Computer Vision (WACV) 2024, pp. 4279-4289

[5] S. Xu, Y. Huang, J. Pan, Z. Ma, and J. Chai, "Inversion-Free Image Editing with Language-Guided Diffusion Models," Proceedings of the IEEE/CVF Conference on Computer Vision and Pattern Recognition (CVPR), 2024, pp. 9452-9461.

**Questions:**

The findings in lines 210-212 are interesting. I am curious whether this phenomenon arises from differences in the text encoder or the diffusion model architecture. Could the authors provide insights into the underlying reasons for this observation?

---

> ### Author Response · Authors · 2024-11-17
> **Thank you for the review!**
>
> We thank the reviewer for providing the feedback. In the following, we respond to each of the concerns in the review. Additionally, the modifications in the revision are marked in blue, for your convenience.
>
> 1. **The proposed techniques are incremental, lacking sufficient novelty to make a significant contribution to the field**
>
>     We appreciate the reviewer's feedback and would like to address the concerns in two parts: first, by clarifying the contributions of our proposed techniques, and second, by highlighting additional core contributions of our work.
>
>     **Techniques Contribution**: The reviewer is correct that some of our method’s components build on prior work. However, even though some techniques have been available for more than a year, object insertion remains a challenging, unsolved problem (as also noted by reviewer 5Sc6). This indicates that our application and refinement of these techniques are non-trivial and advance the state of the art.
>
>     We also highlight that the Weighted-Extended Self-Attention mechanism is based on our novel findings regarding attention distribution within MM-DiT blocks, which the reviewer acknowledged as interesting. While previous methods have explored weighting components like text embeddings [1], our approach uniquely leverages MM-DiT attention sources—a component that has not been explored before. Unlike [1], our weighting parameters are derived from an analytical study of attention dynamics in multi-modal diffusion transformers, contributing to a deeper understanding of models as SD3 and FLUX.
>
>
>     **Further Contributions**: Our work makes additional substantial contributions to the field. **First**, we demonstrate significant improvements in the task of object insertion to images, supported by quantitative metrics, human evaluations, and qualitative examples. **Second**, we address a critical gap in previous object addition methods: ensuring that objects are placed in plausible locations (affordance). By constructing the **Additing Affordance Benchmark**, we revealed that existing methods often fail to insert objects appropriately. This benchmark underscores the field's current limitations, while our method, Add-it, achieves an affordance score of **82.8**, significantly outperforming the best previous method at **47.4**. **Finally**, we present the first analysis of attention sources in the MM-DiT block and leverage this understanding to balance information from both the source and target images.
>
>     [1] https://huggingface.co/docs/diffusers/using-diffusers/weighted_prompts#prompt-weighting
>
> 2. **Include a comparison with additional tuning-free methods such as Masactrl [3], Proximal Guidance [4], and InfEdit [5]**
>
>     Per the reviewer's suggestion, we have included comparisons to MasaCtrl and InfEdit in the updated PDF. The authors of the Proximal Guidance work have not released their code, so we were unable to  compare with their method. In **Section A.3** of the updated PDF, we present an evaluation of these baselines along with another baseline requested by Reviewer 5Sc6. Add-it significantly outperforms these baselines, with the most notable improvement in the affordance metric, as shown in **Tables 1 and 3**.
>
> 3. **Incorrect usage of quotation marks on line 237: The word 'choose' is improperly quoted**
>
>     We thank the reviewer for catching this error. It has been corrected in the updated PDF.
>
> 4. **The findings in lines 210-212 (Attention sources distribution in the MM-DiT block) are interesting. Could the authors provide insights into the underlying reasons for this observation?**
>
>     We appreciate the reviewer's positive feedback on our findings. The newly introduced MM-DiT block integrates information from two sources: the text prompt and the generated image patches, simultaneously. By extending this block to also incorporate information from a source image, our method uniquely combines information from three sources within the same layer. In contrast, prior Unet-based diffusion models handle these sources separately—using a cross-attention layer for text prompts and a self-attention layer for image patches. The need to balance these information sources arises specifically from the architecture of the extended MM-DiT block, where all three sources are blended together within a single layer. Finally, in Section A.4 of the appendix, titled “The Role of Positional Encoding,” we further analyze how positional encoding impacts the extended attention between the source and generated images.
>
> We are happy to address any other questions.

---

> > ### Author Response · Authors · 2024-11-21
> > **Follow-up**
> >
> > We sincerely appreciate the time and effort you have dedicated to reviewing our work.
> >
> > We hope you had an opportunity to review our response from November 17.
> > In it, we provided a discussion of the paper's contributions, additional comparisons to tuning-free methods, and insights regarding attention balancing in the MM-DiT block.
> >
> > Are there any other concerns or questions we can help address? We would be happy to provide further clarification.
> >
> > Thank you,
> >
> > The authors

---

> > ### Comment · Reviewer_9iYS · 2024-11-22
> >
> > Thank you for your patient response. Most of my concerns have been resolved, and I appreciate the effort in your response. Therefore, I've decided to raise my score in recognition of your efforts.

---

> > > ### Author Response · Authors · 2024-11-22
> > >
> > > Thank you for considering our rebuttal and adjusting the score!
> > >
> > > We truly appreciate your recognition and feedback. Your review has greatly benefited our work.

---

### Official Review · Reviewer_mir1 · 2024-10-22

**Soundness:** 3
**Presentation:** 3
**Contribution:** 2
**Rating:** 6
**Confidence:** 4

**Summary:**

This paper introduces Add-it, a training-free method leveraging extended attention mechanisms to seamlessly integrate objects into images from text instructions. Besides, it shows a new benchmark for evaluating object placement plausibility.

**Strengths:**

1. The writing is concise, which makes the paper easy to follow and understand.
2. The method is straightforward and innovatively.
3. The experimental results substantiate the effectiveness of the method proposed in this paper.

**Weaknesses:**

1. The newly introduced benchmark's advantages over other benchmarks are not clearly articulated, and there is a lack of comparison with more existing benchmarks, like MagicBrush.
2. The paper lacks comparison with traditional image quality metrics such as FID and IS, which are commonly used to evaluate the quality of generated images in terms of their realism and diversity.
3. Moreover, while a new benchmark is proposed, there seems to be a scarcity of details regarding it in the paper.

**Questions:**

1. Could you please briefly outline the advantages of the benchmark you've proposed over the existing benchmarks? It would be particularly helpful if you could illustrate these advantages with some statistical data.
2. Have you experimented with modifying the input to incorporate multiple target objects, or just edit it? eg. in the second line of the fig 6 in your paper, the prompt change to "The man is holding a shopping bag in the rain." or "The man wears a red shirt".
3. Although your paper focuses on the 'Add-it' method, I'm curious to know if this editing approach could also be adapted to perform operations like 'remove something' from the image.
4. could you please provide more details about Structure Transfer?

---

> ### Author Response · Authors · 2024-11-17
> **Thank you for the review! (response part 1/2)**
>
> We thank the reviewer for providing the feedback. In the following, we respond to each of the concerns in the review. Additionally, the modifications in the revision are marked in blue, for your convenience.
>
> 1. **What are Additing Affordance Benchmark advantages over other benchmarks? Provide more details**
>
>     The core strength of our proposed benchmark, and its advantage over others, is its ability to quantitatively and automatically assess whether an object was added correctly to an existing scene.
>
>     In Lines 88-93, we highlight a significant gap in current object insertion evaluation protocols: the lack of a mechanism to assess whether an object was added to a *plausible location*. Our experiments reveal that existing methods often struggle to find the “right” spot for object placement, which we refer to as affordance. To address this, we introduced the “Additing Affordance Benchmark,” which includes carefully labeled regions indicating plausible insertion areas for each image sample in the dataset.
>
>     We provide information about the benchmark in Section 4, with additional details on its construction and evaluation protocol in **Section A.6**, titled “Additing Affordance Benchmark.” Sample examples are shown in Figure 19.
>
>     Following the reviewer's suggestion, we included **additional discussion in section A.6** of the updated PDF, highlighting the benchmark’s advantages and necessity in the field. We believe this benchmark is a major contribution, as determining plausible insertion locations is crucial for future object insertion methods. Following Reviewer rVMn suggestion, we have expanded the benchmark from 200 to 500 samples.
>
> 2. **Comparison with the MagicBrush benchmark**
>
>     In response to the reviewer's suggestion, we now added comparisons of Add-it with other baselines on the MagicBrush benchmark, as discussed in **Section A.3** of the updated PDF. We include both qualitative and quantitative comparisons, demonstrating that Add-it outperforms all prior methods, including the MagicBrush model itself.
>
> 3. **The paper lacks a comparison with traditional image quality metrics such as FID and IS, which are common in measuring generated images quality.**
>
>     We agree with the reviewer that incorporating additional quality metrics can be valuable. We first explain why standard FID and IS metrics are not the best choice for image editing, and then propose a modification, which we added to the paper.
>
>     Metrics like FID and IS are commonly used in image generation but  they tend to be less meaningful when evaluating image editing methods, and as a result are generally not employed in popular image editing methods and benchmarks [1,2,3,4,5,6]. This is due to two main reasons: First, image editing keeps most of the original image unchanged, making it difficult to quantify the edit influence on the score. For instance, a model that simply refrains from adding any new objects could achieve a near-perfect FID score. Second, FID performs best with datasets containing at least 5,000 samples [7], whereas image editing benchmarks are typically much smaller, often with only a few hundred samples
>
>     With that said, we agree with the reviewer that incorporating additional quality metrics is valuable for the field. To adapt these metrics for the image addition task, we employ an off-the-shelf open vocabulary detection model to identify regions containing the added objects. We crop these regions to create a dataset of object crops, then evaluate KID and IS by comparing the cropped dataset against the original source images. This method allows us to directly assess the quality and diversity of the inserted objects. We opted for KID over FID due to its reliability on smaller datasets [7]. As discussed in **Section A.3** of the updated PDF and shown in **Table 6**, Add-it achieves the best performance in both KID and IS metrics.
>
>     [1] InstructPix2Pix: Learning to Follow Image Editing Instructions
>
>     [2] EraseDraw: Learning to Insert Objects by Erasing Them from Images
>
>     [3] MagicBrush : A Manually Annotated Dataset for Instruction-Guided Image Editing
>
>     [4] Paint by Inpaint: Learning to Add Image Objects by Removing Them First
>
>     [5] Emu Edit: Precise Image Editing via Recognition and Generation Tasks
>
>     [6] Prompt-to-Prompt Image Editing with Cross Attention Control
>
>     [7] Training Generative Adversarial Networks with Limited Data

---

> > ### Author Response · Authors · 2024-11-17
> > **Thank you for the review! (response part 2/2)**
> >
> > 4. **Can the input be modified to add multiple objects, or edit existing objects?**
> >
> >     While we have not explicitly experimented with adding multiple objects in a single edit, this can be easily achieved through a step-by-step generation process, as shown in Figures 1 and 10. Additionally, although editing existing objects is not the primary focus of our paper, our method is capable of such edits. In **Figure 14** of the updated PDF, we demonstrate multiple edits, ranging from changing a person’s hair and eye color to replacing a car with a boat.
> >
> > 5. **Could your approach be adapted to perform operations like 'remove something' from the image?**
> >
> >     We believe that adding an object to an image is inherently harder than removing it. Adding requires identifying a plausible location to seamlessly integrate the new object, whereas removal focuses on identifying an existing object in the scene. We believe that by combining object detection with an inpainting model, potentially based on Add-it, we could extend our approach to handle object removal; Although we feel such a task lies outside the scope of our paper.
> >
> > 6. **Provide more details about Structure Transfer**
> >
> >     The purpose of the Structure Transfer step is to align the layout of the generated image with that of the source image, ensuring that the added object fits naturally within the existing scene. Previous research has shown that the layout of a generated image is largely determined in the early steps of the diffusion process. We leverage this by injecting a highly noisy version of the source image at an early stage in the generation process of the edited image. In Figure 8, we ablate the impact of the Structure Transfer step. As demonstrated, without this step, the edited image might produce a dog that does not conform to the original scene structure, such as generating a dog disproportionately larger than the chair. Incorporating Structure Transfer results in a dog with a size that aligns properly with the scene. Additionally, we analyze the effect of applying Structure Transfer at different early steps of the generation process.
> >
> > We are happy to address any other questions.

---

> > > ### Author Response · Authors · 2024-11-21
> > > **Follow-up**
> > >
> > > We sincerely appreciate the time and effort you have dedicated to reviewing our work.
> > >
> > > We hope you had an opportunity to review our response from November 17.
> > > In it, we included additional discussion on our Affordance benchmark, comparisons with the MagicBrush benchmark, additional image quality metrics, and further responses to your questions.
> > >
> > > Are there any other concerns or questions we can help address? We would be happy to provide further clarification.
> > >
> > > Thank you,
> > >
> > > The authors

---

> > > > ### Comment · Reviewer_mir1 · 2024-11-22
> > > > **Thanks for your response**
> > > >
> > > > Dear authors,
> > > >
> > > > Your answers have effectively resolved my issue. As a result, I have decided to increase my rating. Thank you once again for your excellent work.

---

> > > > > ### Author Response · Authors · 2024-11-22
> > > > >
> > > > > Thank you for considering our rebuttal and adjusting the score!
> > > > >
> > > > > We truly appreciate your recognition and feedback. Your review has greatly benefited our work.

---

### Official Review · Reviewer_5Sc6 · 2024-11-01

**Soundness:** 3
**Presentation:** 2
**Contribution:** 3
**Rating:** 6
**Confidence:** 3

**Summary:**

Add-it leverages pretrained diffusion models to insert objects into images based on text prompts, without requiring any training.
The method extends the multi-modal attention mechanism to incorporate information from the source image, prompt, and generated image, resulting in natural object placement.
Add-it also introduces a novel “Additing Affordance Benchmark” to evaluate object placement plausibility.

**Strengths:**

1. The paper proposes a novel training-free method called Add-it for adding objects to both real and generated images using simple textual prompts. The method doesn't require fine-tuning or training a separate model for object insertion. Instead, it directly leverages the knowledge embedded within pretrained text-to-image diffusion models.

2. Add-it extends the multi-modal attention mechanism of diffusion models to carefully balance information from three key sources: the source image, the text prompt, and the generated image itself.

3. Add-it introduces a novel Subject-Guided Latent Blending mechanism to preserve the fine details of the source image while enabling necessary adjustments like shadows and reflections.

**Weaknesses:**

1. Lack of comparison with the existing methods for adding objects [1,2]. Both of them have codes and have been released half a year ago.

2. The paper doesn't fully discuss the potential limitations of relying on SAM-2 for refining the object mask. SAM is a large powerful segmentation model, might not always accurately capture the desired object boundaries. There are also papers using self-attention and cross-attention maps to relieve this requirement [3,4].

3. The paper only briefly mentions potential biases inherited from the pretrained diffusion models. These biases could impact the object placement in unexpected ways. For instance, the model might consistently place objects associated with certain demographics in specific locations based on learned biases, potentially leading to unfair or discriminatory outcomes.

[1] Paint by Inpaint: Learning to Add Image Objects by Removing Them First
[2] ObjectAdd: Adding Objects into Image via a Training-Free Diffusion Modification Fashion
[3] Localizing Object-level Shape Variations with Text-to-Image Diffusion Models
[4] Dynamic Prompt Learning: Addressing Cross-Attention Leakage for Text-Based Image Editing

**Questions:**

Please refer to the weakness. Anyway, from my view as a Diffusion Models researcher, adding objects to images is a pretty hard task. Although this paper is having several limitations, I still like the basic idea.

---

> ### Author Response · Authors · 2024-11-17
> **Thank you for the review!**
>
> We thank the reviewer for the positive and comprehensive feedback. In the following, we provide a response to each of the concerns in the review. Additionally, the modifications in the revision are marked in blue, for your convenience.
>
> 1. **Lack of comparison with the existing methods for adding objects [1,2].
> [1] Paint by Inpaint: Learning to Add Image Objects by Removing Them First
> [2] ObjectAdd: Adding Objects into Image via a Training-Free Diffusion Modification Fashion**
>
>     We appreciate the reviewer for highlighting these additional baselines. Regarding baseline [1] (PaintByInpaint), In **Section A.3** of the updated PDF we present an evaluation of [1] along with two other baselines requested by Reviewer 9iYS. Our Add-it significantly outperforms these three baselines, with the most notable improvement in the affordance metric, as shown in **Tables 1 and 3**. Add-it achieves a score of **0.828**, whereas the highest score among these baselines is **0.366**.
>
>
>     Regarding baseline [2] (ObjectAdd). This paper addresses a different task than ours. There, the user specifies the region for object addition as input, making it more akin to an inpainting task. In our paper, the location of the added object is found by the model, not given. This difference prevents a direct comparison with the other baselines. Additionally, the official code released for [2] is missing crucial files, which prevents us from running it.
>
> 2. **Potential limitations of relying on SAM-2 for refining the object mask. Previous work uses self- and cross-attention maps to relieve this requirement**
>
>     Following this comment we added discussion in In **Section A.5** of the updated PDF. We discuss various methods for localizing the added object during the Latent Blending step. We provide both quantitative (**Table 7**) and qualitative (**Figure 18**) comparisons of these alternatives, specifically coarse attention masks and masked-based SAM-2 prompt. Our proposed point-based SAM-2 prompting approach outperforms both alternatives.
>
>     While it is true that SAM-2 may not always perfectly capture object boundaries, in our experiments it consistently outperforms coarse attention-based masks. Note that we do leverage self- and cross-attention maps to generate point-based prompts for SAM-2.
>
> 3. **Potential biases inherited from the pretrained diffusion models**
>
>     We acknowledge the risk of potential biases and have addressed this in the Limitations section (Section 6). However, these biases are not unique to our method; they are common across all current editing approaches that rely on pretrained diffusion models. We included qualitative examples illustrating these biases in **Figure 15** of the updated PDF, and discuss it further in **section A.2**.
>
> We are happy to address any other questions.

---

> > ### Author Response · Authors · 2024-11-21
> > **Follow-up**
> >
> > We sincerely appreciate the time and effort you have dedicated to reviewing our work.
> >
> > We hope you had an opportunity to review our response from November 17.
> > In it, we provided extended baseline comparisons, additional ablations for alternatives to our proposed points-based SAM2 algorithm, and additional discussion and results concerning model biases.
> >
> > Are there any other concerns or questions we can help address? We would be happy to provide further clarification.
> >
> > Thank you,
> >
> > The authors

---

> > ### Comment · Reviewer_5Sc6 · 2024-11-22
> > **Thanks for your rebuttal**
> >
> > My concerns are mostly solved. I will keep my acceptance rating. Thanks!

---

### Official Review · Reviewer_rVMn · 2024-11-04

**Soundness:** 3
**Presentation:** 3
**Contribution:** 3
**Rating:** 6
**Confidence:** 4

**Summary:**

This paper introduces Add-it, a novel, training-free method for inserting objects into images based on text instructions. The approach extends multi-modal self-attention mechanisms in Diffusion Transformer to incorporate a source image as an additional input, reweighting the attention values to effectively balance information between source and target images. Furthermore, the method transfers structural details from the source image and blends the latent representation with guidance from SAM2. To validate Add-it, the authors propose a new benchmark and a custom evaluation metric, demonstrating its effectiveness.

**Strengths:**

- **Clarity and Presentation.** The paper is well-written and easy to follow. The figures are thoughtfully designed and effectively illustrate the proposed method.
- **Visual Quality.**  The results are visually compelling; the added objects are integrated seamlessly, with the original image's details preserved well.

**Weaknesses:**

- **Dataset Size.** The evaluation affordance dataset is relatively small, containing only 200 images, which may limit the generalizability of the findings.

- **Computation Speed.** The need to run additional models (e.g., SAM2) for mask estimation could impact processing speed, especially in comparison to the original FLUX. It would be valuable to provide a direct comparison regarding computational efficiency.

- **Prompt Complexity.** *Add-it* requires text prompts for target images and thus may introduce additional complexity. Previous methods like InstructPix2Pix or MagicBrush may offer a more straightforward user experience.

**Questions:**

- Did the authors experiment with using a coarse mask as an input for SAM2? If so, how does it perform compared to point-based prompts?
- In the ablation study, the authors claim that setting \gama = auto yields the best results. However, Figure 7(a) appears to show a peak at approximately 1.07, which surpasses the performance of the auto setting.

---

> ### Author Response · Authors · 2024-11-17
> **Thank you for the review!**
>
> We thank the reviewer for the positive and comprehensive feedback. In the following, we respond to each of the concerns in the review. Additionally, the modifications in the revision are marked in blue, for your convenience.
>
> 1. **The evaluation affordance dataset is relatively small, containing only 200 images**
>
>     Following this feedback, and to strengthen our benchmark, we have added 300 additional examples, increasing the dataset to a total of 500 images. The newly added examples have been included in the updated supplementary material.
>
>     With this increase, our benchmark now has 2.5x more object insertion samples than in other popular benchmarks such as EmuEdit and MagicBrush (only 200 for the adding task).
>
>
> 2. **Computation Speed, running SAM2 could impact processing speed**
>
>     SAM2 is a real-time video segmentation model, and its computational time is minimal compared to the runtime of the diffusion model. In our experiments, SAM2 inference takes 0.04 seconds on average, whereas FLUX requires 6.4 ± 0.13 seconds per image. Overall, Add-it inference takes about 7.225 ± 0.05 seconds per image—only one second longer than the original FLUX model (both statistics were derived from 200 generations). This slight increase is due to caching attention maps and the additional computation for the extended attention mechanism, with SAM2 contributing only a negligible portion to this difference. We now included statistics on computation time in **Section A.1** of the updated PDF.
>
> 3. **Add-it text prompts may introduce additional complexity over instruction based methods**
>
>     The text prompts used in Add-it are generally quite simple, as demonstrated in Figures 1, 5, 6, and 11. These prompts are similar in structure to instruction-based prompts, making the transition between the two straightforward. For instance, given an input image with the caption “A skateboarder with no shirt doing tricks on a rail” and instruction “Add a football jersey to the torso of the skateboarder”, the Add-it style prompt would be “A skateboarder with a football jersey doing tricks on a rail”. We find that  an LLM can easily translate between these formats, as we demonstrated by using ChatGPT to convert instructions into Add-it style prompts for the EmuEdit dataset. We included further details on this process in **Section A.7** of our updated PDF.
>
> 4. **Did the authors experiment with using a coarse mask as an input for SAM2?**
>
>     Yes, we initially experimented with using the coarse attention mask as input for SAM2. However, we found that it generally performs worse than our proposed point-based algorithm. Following the reviewer's question, we have included a discussion in **Section A.5** of the updated PDF concerning alternative methods for localizing the added object during the Latent Blending step. We provide both quantitative (**Table 7**) and qualitative (**Figure 18**) comparisons of these alternatives, specifically (i) coarse attention masks, and (ii) masked-based SAM-2 prompts. Our proposed point-based SAM-2 prompting approach outperforms both alternatives, with an Affordance score of **0.828** compared to scores of 0.77 and 0.809 for the alternatives.
>
> 5. **Figure 7(a) shows peak at gamma=1.07, surpassing the gama=auto setting**
>
>     You are correct. The scale value we used in our experiments is gamma = 1.05. While using gamma=auto will usually yield better results. We opted for a fixed value for simplicity and efficiency. The fixed value was determined by averaging the auto-gamma values that balanced the attention sources on a separate validation set. To avoid test leakage, we did not determine a new optimal gamma for each evaluation set individually. However, even with the fixed gamma optimized for our validation set, Add-it still significantly outperforms all previous methods. While selecting gamma=1.07 could improve results slightly, it would risk introducing test leakage.
>
> We are happy to address any other questions.

---

> > ### Author Response · Authors · 2024-11-21
> > **Follow-up**
> >
> > We sincerely appreciate the time and effort you have dedicated to reviewing our work.
> >
> > We hope you had an opportunity to review our response from November 17.
> > In it, we included an extended benchmark size, additional details on computation speed, and further ablations for alternatives to our proposed points-based SAM2 algorithm.
> >
> > Are there any other concerns or questions we can help address? We would be happy to provide further clarification.
> >
> >
> > Thank you,
> >
> > The authors

---

> > > ### Comment · Reviewer_rVMn · 2024-11-23
> > > **Thanks for your response**
> > >
> > > Thank you for the authors’ response. Most of my concerns have been addressed, and I will maintain my current rating.

---

### Author Response · Authors · 2024-11-17
**General response to reviewers**

We thank all the reviewers for providing useful and insightful feedback. We are thrilled that the reviewers found our work to be **novel and innovative** (Reviewer rVMn, Reviewer 5Sc6, Reviewer mir1), our **state-of-the-art** results **compelling and natural** (Reviewer rVMn, Reviewer 5Sc6, Reviewer mir1, Reviewer 9iYS), our MM-DiT attention-balancing analysis **interesting and effective** (Reviewer rVMn,Reviewer 9iYS) and acknowledged the **merit** of our proposed “Additing Affordance Benchmark” (Reviewer rVMn, Reviewer 5Sc6, Reviewer 9iYS).

Our work has benefited tremendously from your feedback. Below are the main modifications to the manuscript (colored blue in the updated PDF):
1. **Comparisons:** Following the suggestions by Reviewers 5Sc6, mir1 and 9iYS, we added comparisons with three additional baselines: Paint by Inpaint, InfEdit, and MasaCtrl. Additionally, we added evaluations on the MagicBrush benchmark. We also introduced two metrics, the KID and IS to quantify the quality and diversity of added objects.
2. **Additing Affordance Benchmark:** Following the suggestions by Reviewers rVMn and mir1, we expanded the Additing Affordance Benchmark from 200 to 500 samples, each including source images, edit instructions, and manually annotated bounding boxes indicating plausible insertion regions. We also expanded the discussion of the Additing Affordance Benchmark, providing details about its guiding principles, contributions, and advantages over existing benchmarks.
3. **Additional Results:** Following the suggestions by Reviewers 5Sc6 and mir1, we added results demonstrating Add-it’s capability to edit existing objects, as well as an analysis of potential biases inherited from the pre-trained diffusion model.
4. **Ablations:**  Following the suggestions by Reviewers rVMn and 5Sc6, we conducted ablations on alternatives for the Latent Blending mask construction procedure. Specifically, we added both quantitative and qualitative evaluations of alternatives to the points-based SAM2 prompts, including using coarse-attention masks and mask-based SAM2 prompts.

We look forward to addressing any remaining questions from the reviewers and continue engaging in further discussion. If you find our response satisfactory, we kindly ask you to consider raising your rating in recognition of our core contributions, namely: a state-of-the-art method that significantly advances the challenging task of object insertion, the first evaluation of attention sources in the MM-DiT block, and the introduction of an affordance benchmark that addresses a critical gap in current image addition methods and evaluation protocols.

---

### Author Response · Authors · 2024-11-27

Thank you to all reviewers for dedicating your time to review our paper, and for your continuous engagement during the discussion period.

We are pleased to have addressed all your concerns and are delighted to see that all reviewers support the acceptance of our paper to ICLR.

Given the extension of the discussion period, we would greatly appreciate any further feedback, so we can strengthen the paper and allow the committee to make a more confident decision.

---

### Meta-Review · Area_Chair_vuD6 · 2024-12-19

**Metareview:**

The paper proposes a training-free approach to insert new objects into images based on text prompts, with a pretrained diffusion model. Reviewers acknowledge several strengths of the paper, including the good presentation, effectiveness of the proposed approach validated in experiments, and the novel and straightforward idea. The weaknesses proposed by reviewers include lack of comparison to other baseline approaches and other benchmarks, additional results, analysis, and ablations. The authors addressed these questions in the rebuttal and all reviewers give a positive score of 6. Area chair agree with reviewers and would recommend accepting this paper as poster.

**Additional Comments On Reviewer Discussion:**

Most reviewer questions focus on experiments, including comparison to other approaches and other benchmarks, ablation studies, additional results and analysis. One reviewer also mentioned the lack of novelty. The authors addressed these questions in the rebuttal and all reviewers decided to keep their positive scores or increase their score to positive.

---

### Decision · Program_Chairs · 2025-01-22

Accept (Poster)